# Beta rhythmicity in human motor cortex reflects neural population coupling that modulates subsequent finger coordination stability

Seitaro Iwama [1,2], Takufumi Yanagisawa [3,4,5,6], Ryotaro Hirose[1] & Junichi Ushiba [7✉]

Human behavior is not performed completely as desired, but is influenced by the inherent rhythmicity of the brain. Here we show that anti-phase bimanual coordination stability is regulated by the dynamics of pre-movement neural oscillations in bi-hemispheric primary motor cortices (M1) and supplementary motor area (SMA). In experiment 1, pre-movement bi-hemispheric M1 phase synchrony in beta-band (M1-M1 phase synchrony) was online estimated from 129-channel scalp electroencephalograms. Anti-phase bimanual tapping preceded by lower M1-M1 phase synchrony exhibited significantly longer duration than tapping preceded by higher M1-M1 phase synchrony. Further, the inter-individual variability of duration was explained by the interaction of pre-movement activities within the motor network; lower M1-M1 phase synchrony and spectral power at SMA were associated with longer duration. The necessity of cortical interaction for anti-phase maintenance was revealed by sham-controlled repetitive transcranial magnetic stimulation over SMA in another experiment. Our results demonstrate that pre-movement cortical oscillatory coupling within the motor network unknowingly influences bimanual coordination performance in humans after consolidation, suggesting the feasibility of augmenting human motor ability by covertly monitoring preparatory neural dynamics.

[1] School of Fundamental Science and Technology, Graduate School of Keio University, Kanagawa, Japan. [2] Japan Society for the Promotion of Science, Tokyo, Japan. [3] Institute for Advanced Co-Creation Studies, Osaka University, Osaka, Japan. [4] Department of Neurosurgery, Osaka University Graduate School of Medicine, Osaka, Japan. [5] Department of Neuromodulation and Neurosurgery, Osaka University Graduate School of Medicine, Osaka, Japan. [6] Department of Neuroinformatics, ATR Computational Neuroscience Laboratories, Kyoto, Japan. [7] Department of Biosciences and Informatics, Faculty of Science and Technology, Keio University, Kanagawa, Japan. ✉email: ushiba@bio.keio.ac.jp

Humans often experience situations requiring bilateral finger coordination in daily life, such as typing words on a keyboard or playing the piano. However, performance fluctuates even among repetitions of the same movements[1–5]. This instability of human behavior has been partly ascribed to rhythmicity of the brain. For instance, amplitude of neural oscillations in the alpha-band (8–13 Hz) over the visual cortices causally influences the variability of perception[6–8]. In the motor domain, neural oscillations in the beta-band (14–30 Hz) over sensorimotor cortices, termed sensorimotor rhythm (SMR), are linked to the speed of movement initiation and cancellation[9–11]. Controlling the rhythmicity of the brain might improve the performance of body movements and perception.

The performance of bimanual coordination tasks, which require individualized movement of bilateral fingers, is influenced by rhythmic activities over a distributed motor network, comprising primary motor cortices (M1) and supplementary motor area (SMA)[12–19]. In particular, previous studies have demonstrated that inter-areal phase coupling of neural oscillations in the motor network is associated with bimanual motor behavior[14,16–18,20]. For instance, entrainment of activities within the motor network externally induced by brain stimulation causes errors in the production of sequential bimanual finger movements[21,22]. In addition, functional neuroimaging techniques, including scalp electroencephalograms (EEGs), revealed that bi-hemispheric M1 and SMA are activated and interact with each other during unstable anti-phase finger coordination (Fig. 1a)[13,14,23,24]. Task-related bi-hemispheric phase coupling of SMR during anti-phase finger tapping is stronger than that of in-phase finger tapping[14]. Collectively, task-related cortical interaction within the motor network may underlie stable bimanual coordination by regulating the large-scale neuronal communication with spatiotemporally organized phase synchrony[25,26].

However, there is no direct evidence that manipulation of movement onset dependent on fluctuating cortical coupling among bilateral M1 and SMA modulates subsequent performance of bimanual coordination. If coupling and decoupling of the interareal rhythmic activities represented by neural oscillations are major contributing factors for the performance of bimanual coordination, the quality of performance could be regulated unbeknownst to participants by initiating a motor task depending on the SMR signals in the motor network. Herein, we hypothesized that manipulating movement cues based on pre-movement phase coupling of bi-hemispheric SMR signals might influence the stability of subsequent anti-phase bimanual coordination, which spontaneously transits into the in-phase due to motor command interference[13,27,28].

To test our hypothesis, we used two manipulative approaches. First, we instructed participants to perform an anti-phase tapping task, depending on bilateral SMR phase synchrony estimated online, to demonstrate that the manipulation of movement cues based on pre-movement M1-M1 phase synchrony modulates the duration of anti-phase bimanual tapping (Fig. 1b). Second, we investigated whether upstream of M1 is necessary to modulate M1-M1 coupling and behavioral performance improvement. In particular, SMA was perturbed by repetitive transcranial magnetic stimulation (rTMS). Experiment 1, which used brain state-dependent movement initiation, revealed whether the manipulation of movement cues based on pre-movement M1-M1 phase synchrony improves motor performance. In comparison, in experiment 2, external neural perturbation by rTMS over SMA would alter the cortical interaction with SMA as a hub, and allow us to evaluate its necessity for the performance of bimanual coordination.

## Results

**Task description**. We investigated whether pre-movement M1-M1 phase synchrony influences subsequent performance on the anti-phase bimanual tapping task using index and middle fingers (Fig. 1a)[13,14,29,30]. Since anti-phase finger tapping spontaneously transits to in-phase tapping (Supplementary Movie 1), its duration represents the capability of maintaining unstable bimanual coordination. The timing of phase transition was identified when the tapping time difference of the anti-phase pair (e.g., right index

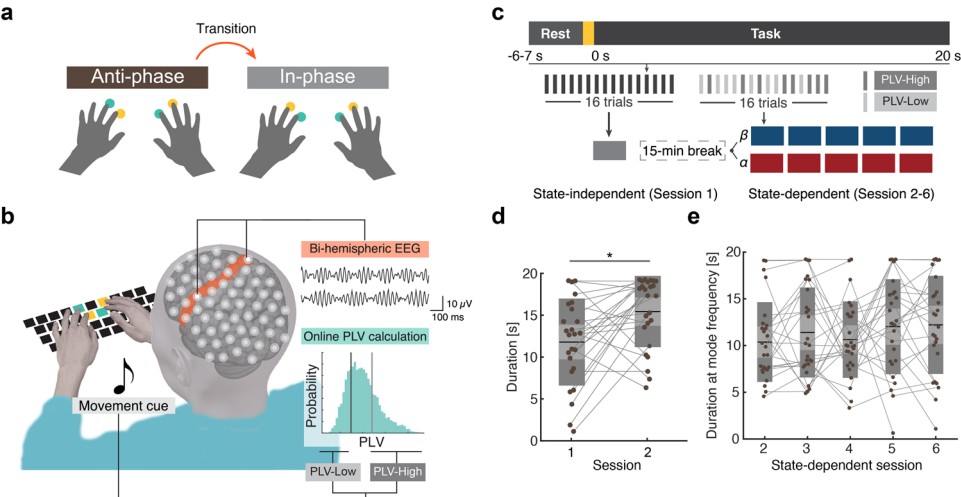

**Fig. 1 Anti-phase bimanual tapping task. a** Schematic of the behavioral task. **b** Schematic of the experimental setup. Online acquired scalp EEG signals were processed online to calculate phase-locking values (PLVs) of bi-hemispheric SMR signals. Movement cues were presented if the PLV satisfied PLV-High or PLV-Low conditions. **c** Structure of experiment 1. *Top*: Time-course of anti-phase bimanual tapping task. The yellow area indicates the ready period whose duration was variable. *Bottom*: Schematic of the experimental procedure. Participants were allocated to the beta- or alpha-group after a state-independent session and 15-min break. In both groups, participants underwent 5 state-dependent sessions. **d** Changes in the duration of anti-phase bimanual tapping between sessions 1 and 2. The mean duration of trials at mode frequency in the two sessions was compared using a paired *t*-test (*: $p < 0.05$, exact values in the main text). The smaller light and larger dark gray areas represent 95% confidence interval and 1 SD, respectively. The black line represents mean values and points connected with gray lines indicate data from each participant. **e** Changes in duration of anti-phase bimanual tapping among the triggered sessions. The mean duration of trials performed at mode frequency in the five sessions were subjected to one-way ANOVA.

and left middle fingers) exceeds that of the corresponding in-phase pair (right and left index fingers). In the present study, a total of 62 participants performed the anti-phase tapping task paced by an intermittent beep sound during scalp EEG acquisition (Fig. 1b, See Supplementary Fig. 1 for quality assurance of EEG signals). In the first trial, participants underwent a 20-s anti-phase tapping task paced by 2-Hz beep sounds (Fig. 1c). In the next trial, the frequency of the beep sound was increased by +0.5 Hz, if the duration of the anti-phase tapping task exceeded 90% of the task period.

In experiment 1, 30 participants underwent 6 sessions of anti-bimanual finger-tapping tasks (16 trials for each session). The first session was a state-independent session, in which cues for movement initiation (i.e., intermittent beep sound) were presented independently from the scalp EEG signals. On the other hand, the remaining sessions were state-dependent, in which online estimated M1-M1 phase synchrony was monitored to inform participants to start tapping depending on the phase synchrony. The phase-locking values (PLVs) between EEG signals recorded from bi-hemispheric motor cortices were calculated online to estimate the interhemispheric M1-M1 phase synchrony for a certain frequency band[31]. Thirty participants were randomly assigned to two groups, in which tapping was initiated based on the PLVs in beta- or alpha-band (beta: 14–30 Hz; alpha: 8–13 Hz). Mode of tapping speed, instructed by the frequency of beep sound, did not exhibit statistically significant difference (Beta-group: $3.95 \pm 0.8$ Hz, Alpha-group: $3.70 \pm 0.6$ Hz, two-sample $t$-test, $t(28) = 0.97$, $p = 0.34$). During state-dependent sessions, participants underwent two conditions to perform the anti-phase bimanual tapping task at high or low pre-movement M1-M1 PLV calculated every 100 ms using the latest 1-s data (Fig.1c, See also"Online signal processing"). The two conditions were pseudo-randomized in each session (8 trials for each condition) and configured unbeknownst to participants. It should be noted that participants performed the task similar to the state-independent sessions, if the pre-movement PLVs did not satisfy the conditions within the ready period.

A 15-min break was interposed between the first and second sessions to allow filter calibration, which may have influenced performance improvement through offline consolidation. Consistent with previous behavioral studies[32–34], the comparison of anti-phase duration in state-independent trials exhibited significant behavioral improvement between sessions 1 and 2 (Fig. 1d, paired t-test, $t(24) = 2.77$, $p = 0.011$, $d = 0.78$, $CI_{95}= [0.20\ 1.35]$), suggesting that the short-term break consolidated anti-phase tapping[35]. It should be noted that the trials in session 2 that were included in the comparison did not satisfy PLV-High or PLV-Low conditions, and were performed independently from M1-M1 PLV. Because the anti-phase duration did not exhibit systematic differences among state-dependent sessions (Fig. 1e, one-way ANOVA, $F(4,126) = 0.79$, $p = 0.53$, Supplementary Fig. 2a, b), the trials derived from each session were aggregated in the following analysis.

**Manipulation of movement cues based on pre-movement phase synchrony of bilateral motor cortices improved the duration of anti-phase bimanual tapping.** We confirmed whether PLV-dependent movement initiation successfully modulated pre-movement M1-M1 PLV. The PLV-dependent paradigm successfully distinguished between the high and low conditions in both groups (Fig. 2a, Supplementary Fig. 1f, paired t-test, beta: $t(14) = 3.84$, $p = 0.0018$, $d = 0.99$, $CI_{95}= [0.356-1.602]$; alpha: $t(14) = 5.95$, $p = 0.00004$, $d = 1.54$, $CI_{95} = [0.767, 2.282]$). However, the duration of anti-phase bimanual coordination was significantly different in the beta-band group but not in the

alpha-band group (Fig. 2b, paired t-test, beta: $t(9) = 2.78$, $p = 0.022$, $d = 0.88$, $CI_{95} = [0.124--1.60]$; alpha: $t(11) = 0.272$, $p = 0.79$, $d = 0.08$, $CI_{95} = [-0.49, 0.64]$). Note that only participants who contained more than 5 trials for each condition and whose pre-movement PLV was successfully conditioned were included in the statistical test (See material and methods for details). In both groups, the pre-movement magnitude of event-related spectral perturbation (ERSP) at the SMA in alpha- and beta-band was not significantly different between the two conditions (Fig. 2c, a mixed two-way repeated measures ANOVA, ERSP in beta-band: main effect of "Time", $F(1, 20) = 0.31$, $p = 0.59$, "Group", $F(1, 20) = 0.25$, $p = 0.62$, interaction of "Time" × "Group", $F(1, 20) = 0.11$, $p = 0.74$; ERSP in alpha-band: main effect of "Time", $F(1, 20) = 0.22$, $p = 0.65$, "Group", $F(1, 20) = 0.27$, $p = 0.61$, interaction of "Time" × "Group", $F(1, 20) = 0.92$, $p = 0.35$). Consistent with the significant difference in duration at the beta-band group, a negative correlation between the pre-movement PLV and duration of anti-phase tapping was significant in the beta-band (Fig. 2d. repeated measures correlation test[36], beta: $r = -0.65$, $p = 0.016$, $CI_{95} = [-0.90, -0.083]$; alpha: $r = -0.29$, $p = 0.33$, $CI_{95} = [-0.76, 0.37]$). In both groups, the sampled trial did not exhibit a significant difference at each condition (Supplementary Fig. 2c, paired t-test, beta-group: $t(12) = -1.15$, $p = 0.273$; alpha-group: $t(14) = -0.06$, $p = 0.95$), thereby excluding temporal bias between conditions due to adaptation or fatigue. Collectively, pre-movement bilateral phase synchrony of SMR modulated the subsequent anti-phase bimanual tapping duration, and M1-M1 PLV was negatively correlated with the duration at the group-level. Notably, during debriefing at the end of the experiment, none of the participants reported that they noticed the existence of two different conditions, which avoided its influence on performance. We found similar results using alternative parameters and metrics for phase coupling applied to the source-space EEG data (Supplementary Fig. 3 and 4a–c), but not for the phase-shuffled surrogate dataset (Supplementary Fig. 4d).

**Pre-movement activity patterns of the motor network explained individual variability of bimanual coordination performance.** The comparison of anti-phase tapping task performance initiated by M1-M1 phase synchrony showed that manipulation of movement cues based on pre-movement M1-M1 PLV modulated subsequent stability of anti-phase bimanual coordination performance. Given that M1-M1 connectivity is modulated by SMA[12], we further investigated the relationship between the individual performance of anti-phase bimanual coordination and activity patterns within the motor network, consisting of SMA and bilateral M1. As shown in Fig. 3a, the pre-movement magnitude of ERSP derived from SMA did not significantly correlate with M1-M1 PLV at the group-level (Pearson's correlation test, beta: $r = 0.21$, $p = 0.50$, $CI_{95} = [-0.41, 0.70]$; alpha: $r = 0.20$, $p = 0.49$, $CI_{95} = [-0.44, 0.71]$). However, pre-movement SMA ERSP magnitude in the beta-band was negatively correlated with the duration of anti-phase tapping (Fig. 3b, Pearson's correlation test, beta: $r = -0.75$, $p = 0.0048$, $CI_{95} = [-0.94, -0.22]$; alpha: $r = -0.453$, $p = 0.10$, $CI_{95} = [-0.82, 0.19]$), suggesting that the pre-movement ERSP magnitude and M1-M1 PLV contributed to the inter-individual variability of anti-phase bimanual coordination[37].

To investigate whether individual variability of bimanual finger coordination performance is explained by the characteristics of the SMA-M1 motor network during the pre-movement period, multiple regression models were constructed to evaluate the effects of M1-M1 PLV, ERSP magnitude at SMA, and their interactions. Their interaction was significant only in the beta-band (Fig. 3c; beta-band: $p = 0.007$ vs. constant model, ERSP × PLV: $\hat{\beta} = 0.74$, $p = 0.003$,

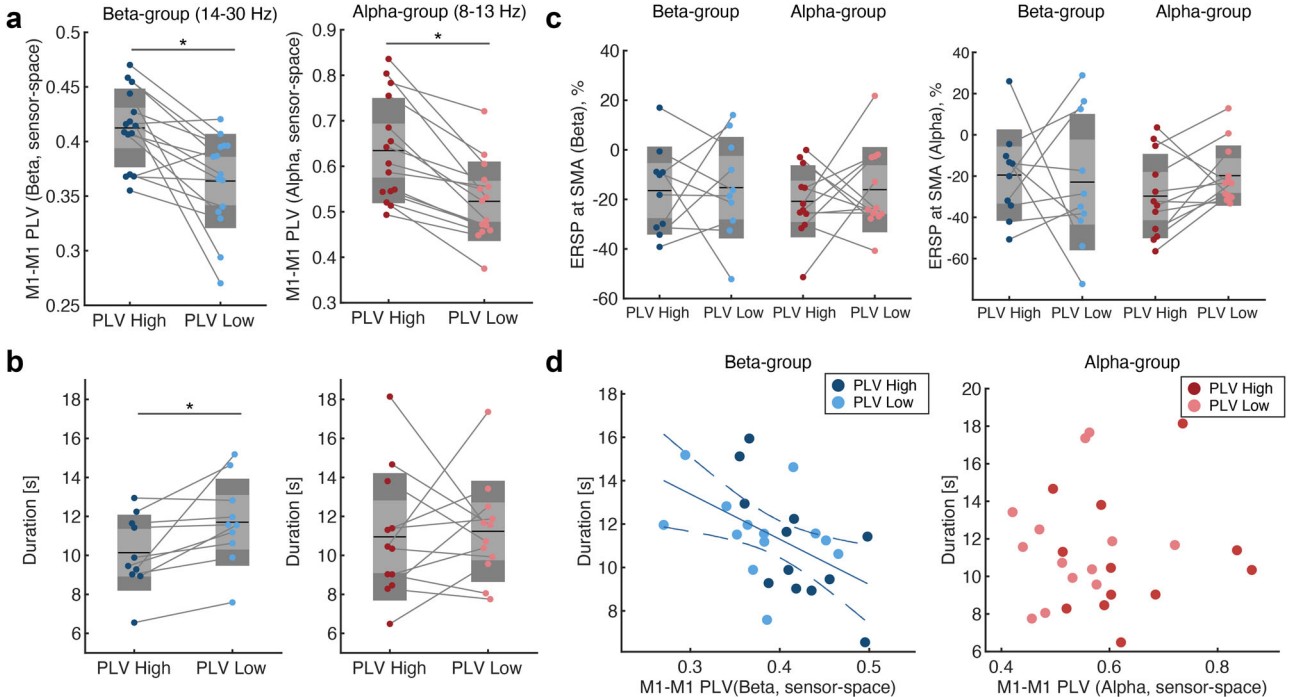

**Fig. 2 Performance improvement of anti-phase bimanual coordination using brain state-dependent movement initiation paradigm. a** Comparison of phase-locking values (PLVs) of bilateral primary motor cortices (M1) between two conditions. Groups based on M1-M1 PLVs in both beta- and alpha-bands exhibited significant differences in high and low conditions (paired t-test, *p < 0.05, exact values in the main text). **b** Comparison of duration of anti-phase bimanual tapping between the two conditions. The group based on M1-M1 PLV in the beta-band exhibited significant differences between high and low conditions (paired *t*-test, *p < 0.05, exact values in the main text). **c** Comparison of pre-movement ERSP at SMA. *Left*: ERSP in the beta-band (a mixed two-way repeated measures ANOVA, statistical values in the main text). *Right*: ERSP in the alpha-band (a mixed two-way repeated measures ANOVA, statistical values in the main text). **d** Correlation analysis between M1-M1 PLV and duration (repeated measures correlation test, beta: r = −0.65, p = 0.016; alpha: r = −0.29, p = 0.33). Solid and dotted lines indicate linear regression and a 95% confidence interval, respectively.

ERSP: $\hat{\beta}=$ −0.468, $p = 0.002$, M1-M1 PLV: $\hat{\beta}=$ −4.15, $p = 0.61$; alpha-band: $p = 0.29$ (vs. constant model)), suggesting that the interplay of higher excitability of SMA and lower M1-M1 PLV collectively contributed to the anti-phase duration.

**Neural perturbation over SMA prevented stabilization of bimanual coordination and reorganization of cortical interaction.** In experiment 1, the pre-movement neural activity patterns, represented by M1-M1 PLV and ERSP magnitude at SMA, explained the inter-individual variability of behavioral performance. If SMA activities organize M1-M1 phase synchrony before the movement occurs, cortical interaction would be the key to the stabilization of subsequent bimanual coordination. To test the necessity of SMA dynamics for stable anti-phase bimanual coordination, the experiment 2 involved offline 1-Hz rTMS treatment or sham stimulation of another 32 participants, followed by pre- and post-evaluation of anti-phase bimanual finger tapping (See methods for speed adjustment). Since a short-term break consolidated the anti-phase tapping task and led to improved performance (Fig. 1d), rTMS perturbation would inhibit the process if SMA plays a critical role in the reorganization of cortical interaction underlying motor control. We selected active sham-stimulation as a control condition due to the absence of a suitable stimulation site for the bimanual motor task whose performance can be influenced by neural substrates for sensorimotor, perceptual as well as cognitive functions[38,39]. The comparison of treatment and active sham stimulation group would reveal the stimulation influences on offline consolidation by controlling the potential burden on coil weight and stimulation-induced twitches[40,41].

First, we evaluated the effect of rTMS treatment on EEG signals during the pre-movement period. Although beta-band ERSP magnitude at SMA did not exhibit any significant changes (Fig. 4a, a mixed two-way repeated measures ANOVA, interaction of "Time" ×"Group", $F(1, 30) = 0.25$, $p = 0.62$, main effect: "Time": $F(1, 30) = 0.031$, $p = 0.86$, "Group": $F(1,30) = 4.02$, $p = 0.054$), pre-movement M1-M1 PLV exhibited significant effects of interaction of "Time" ×"Group" (Fig. 4b, a mixed two-way repeated measures ANOVA, $(1, 28) = 4.94$, $p = 0.035$, $\eta^2 = 0.077$, main effect of "Time", $(1, 28) = 6.73$, $p = 0.015$, $\eta^2 = 0.11$), but not a main effect of "Group" $((1, 28) = 1.18$, $p = 0.29$). In post hoc $t$-tests, a significant decrease in M1-M1 PLV was observed in the sham group (paired t-test, $t(15) = 3.09$, $p = 0.004$, $d = 0.92$, $CI_{95} = [0.32, 1.50]$), but not the treatment group ($t(15) = 0.96$, $p = 0.35$). The baseline M1-M1 PLV did not exhibit significant differences between the two groups ($t$-test, $t(30) = −1.04$, $p = 0.307$). Therefore, it was suggested that rTMS treatment inhibited the reorganization of pre-movement M1-M1 phase synchrony by perturbing offline consolidation at SMA[41]. We found similar results using alternative parameters to calculate M1-M1 PLV (Supplementary Fig. 5).

Accordingly, we investigated the changes in the duration of anti-phase bimanual tapping. A mixed two-way repeated measures ANOVA revealed significant interaction of "Time" × "Group" effect (Fig. 4c, F(1, 30) = 5.24, $p = 0.029$, $\eta^2 = 0.12$) and main effect of "Time" ($F(1, 30) = 9.38$, $p = 0.005$, $\eta^2 = 0.21$), but no main effect of "Group" ($F(1, 30) = 1.59$, $p = 0.23$). The post hoc paired t-test revealed group-specific improvement in anti-phase duration in the sham group but not treatment group (treatment: $t(15) = −0.76$, $p = 0.46$; sham: $t(15) = −3.11$, $p = 0.007$, $d = 0.89$, $CI_{95} = [0.21, 1.33]$). Meanwhile, the baseline

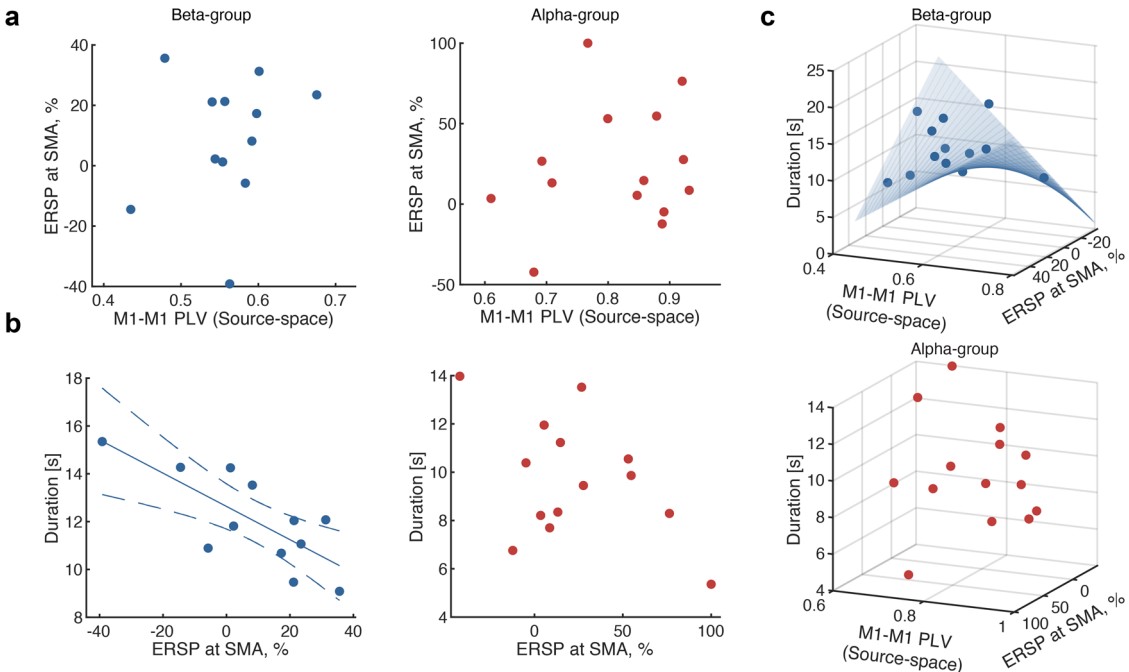

**Fig. 3 Pre-movement cortical interaction explains the behavioral performance. a** Correlation analysis of M1-M1 phase synchrony and event-related spectral perturbation (ERSP) at SMA. No significant correlation was observed in beta and alpha groups (Pearson's correlation test, beta: $r = 0.21$, $p = 0.50$; alpha: $r = 0.20$, $p = 0.49$). **b** Correlation analysis of ERSP at SMA and duration of anti-phase bimanual tapping. Significant correlation was only observed in the beta-group (Pearson's correlation test, beta: $r = -0.75$, $p = 0.0048$; alpha: $r = -0.45$, $p = 0.10$). **c** Multiple regression analysis for the performance and pre-movement cortical activity patterns. M1-M1 PLV, ERSP at SMA, and their interaction were significantly associated with performance only in the beta-band (*Top*: beta-band: $p = 0.007$ (vs. constant model); *Bottom*: alpha-band: $p = 0.29$ (vs. constant model), statistical values in the main text). The blue curved surface represents the regression model.

duration did not exhibit a significant difference between the groups (two-sample t-test: (30) = −0.33, $p = 0.74$).

To test whether transient neural perturbation over SMA modulated the sensorimotor control, we calculated the variability of inter-tap intervals, which indicates how participants tapped the key in keeping with the presented sound. No group-specific changes were observed during anti-phase, in-phase, or unilateral finger tapping tasks (Supplementary Fig. 6, a mixed two-way repeated measures ANOVA, interaction of "Time" × "Group", anti-phase: $F(1, 30) = 1.06$, $p = 0.31$; in-phase: $F(1, 30) = 0.48$, $p = 0.50$; unilateral: $F(1, 28) = 0.04$, $p = 0.85$). These behavioral results are in keeping with studies of rTMS stimulation of SMA[22,42].

Given that rTMS specifically impaired the performance improvement in anti-phase bimanual coordination, whilst no systematic ERSP modulation was observed in SMA signals, we tested whether the association between pre-movement ERSP magnitude at SMA and behavioral performance is preserved after stimulation. To this end, pre- and post-stimulation data from each group were independently subjected to correlation analysis. As shown in Supplementary Fig. 7, the duration of anti-phase bimanual tapping correlated with pre-movement beta-band ERSP magnitude of SMA in the post-treatment data of both groups (Spearman's correlation test, treatment: $\rho = -0.624$, $p = 0.012$, $CI_{95} = [-0.85, -0.19]$; sham: $\rho = 0.756$, $p = 0.001$, $CI_{95} = [-0.91, -0.42]$), but not that of pre-treatment in both groups (treatment: $\rho = -0.418$, $p = 0.11$, $CI_{95} = [-0.78, 0.10]$; sham: $\rho = -0.297$, $p = 0.26$, $CI_{95} = [-0.69, 0.23]$). The cross-experiment replication (Fig. 3b for experiment 1) suggests that pre-movement SMA responses represented by ERSP magnitude in beta-band were involved in bimanual coordination performance even after a short-term break with neural perturbation. In keeping with this finding, differences in beta-band ERSP

magnitude at SMA between pre- and post-stimulation were significantly correlated with the anti-phase duration in the treatment group (Fig. 4d, Spearman's rank correlation test, beta: $\rho = -0.541$, $p = 0.033$, $CI_{95} = [-0.75, -0.17]$), indicating that the variability in behavioral changes by rTMS treatment can be partly explained by the modulation of pre-movement ERSP magnitude at SMA. We found no statistically significant group-specific differences in ERSP magnitude in other regions or frequency of interest (Supplementary Figs. 8, 9a, b) or association with behavioral performance (Supplementary Fig. 9c, d).

Finally, we tested whether neural perturbation over SMA affected the relationship between pre-movement cortical activity patterns and inter-individual variability of anti-phase bimanual tapping duration. A linear-mixed model based on ERSP at SMA, M1-M1 PLV, and their interactions revealed that the interaction with M1-M1 PLV and ERSP at SMA was only significant in the sham group but not in the treatment group (Fig. 4e: treatment: ERSP, $p = 0.01$, PLV, $p < 0.001$, ERSP × PLV: $p = 0.72$; sham: ERSP, $p = 0.038$, PLV, $p = 0.04$; ERSP × PLV: $p = 0.03$), suggesting that rTMS treatment disrupted the effect of cortical interaction between SMA and bilateral M1 phase synchrony on behavioral performance. Meanwhile, the multiple regression model for the alpha-band did not exhibit significant interaction effects in treatment or sham groups (Supplementary Fig. 10).

## Discussion

Our results demonstrate that pre-movement cortical interaction in the motor network modulates subsequent bimanual coordination performance. In particular, we showed that the duration of anti-phase bimanual tapping was prolonged when participants started the task with lower pre-movement M1-M1 PLV (Fig. 2). The interaction of M1-M1 PLV and ERSP magnitude of SMA explained the inter-individual variability of duration after the

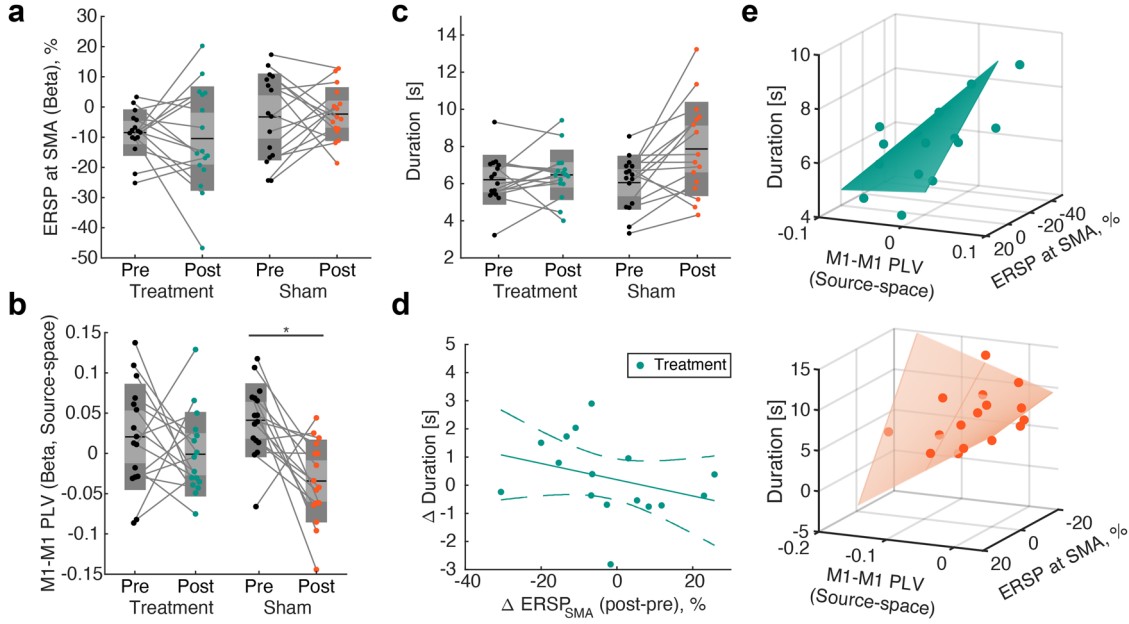

**Fig. 4 Treatment effects of repetitive transcranial magnetic stimulation (rTMS). a** ERSP magnitude at SMA in the beta-band at pre- and post-evaluation of anti-phase bimanual coordination. A mixed two-way repeated measures ANOVA did not exhibit any significant interaction or main effect (statistical values in the main text). **b** Pre-movement M1-M1 PLVs in the beta-band. Differences in phase shuffled data were subjected to a mixed two-way repeated measures ANOVA. A significant interaction of "Time" ×"Group" was observed ($p = 0.035$, statistical values in the main text). A significant difference in the sham group revealed by post hoc $t$-tests (*:$p < 0.05$, statistical values in the main text). **c** Duration of anti-phase bimanual tapping in pre- and post-evaluation sessions. A mixed two-way repeated measures ANOVA revealed significant interaction of "Time" × "Group" effect ($p = 0.029$), and post hoc $t$-test indicated significant difference in the sham group (*:$p < 0.05$, statistical values in the main text). **d** Correlation analysis of changes in duration and SMA-ERSP magnitude in the beta band in the treatment group. Spearman's rank correlation test revealed significant correlation ($\rho = -0.54$, $p = 0.033$). Solid and dotted lines indicate linear regression and a 95% confidence interval, respectively. **e** Linear-mixed model for the performance and pre-movement cortical activity patterns in beta-band. In line with the data from experiment 1 (Fig. 3), a significant main effect was observed for M1-M1 PLV and ERSP magnitude at SMA, and their interaction revealed significant interaction of SMA-ERSP × M1-M1 PLV effect only in sham treatment group (treatment: ERSP: $p = 0.01$, PLV: $p < 0.001$, ERSP × PLV: $p = 0.72$; Sham: ERSP: $p = 0.038$, PLV: $p = 0.04$, ERSP × PLV: $p = 0.03$). The green and orange curved surfaces represent regression model derived from the treatment and sham groups, respectively.

consolidation of the finger tapping task (Fig. 3). Moreover, the effects of the interaction and performance improvement were disrupted by offline perturbation of SMA with rTMS treatment (Fig. 4).

Neural substrates engaged in bimanual coordination have been extensively studied over the last few decades[12–14,17,18,20,21,23,24,27,43–46]. Although operant conditioning of bi-hemispheric phase coupling modulates the speed of bimanual movements after training[20], it is unclear how motor preparation modulates rhythmic interaction within the motor network, which influences subsequent behavioral performance. The main finding of the present study related to bimanual motor control is the demonstration of the functional significance of unconsciously fluctuating intrinsic activities within the motor network, manifested by the causal association with subsequent behavioral performance in the two independent experiments targeting bi-hemispheric M1 phase synchrony and its upstream.

During sequential behavior, SMA plays a critical role in producing motor commands and organizing the downstream motor network, such as bilateral M1[12,29,47,48]. We found that their oscillatory activities during motor preparation organize the motor network for subsequent performance improvement after consolidation. Since the anti-phase bimanual tapping task requires bilaterally independent, yet coordinated, activation of cortico-motor pathways (e.g., simultaneous activation of the left index and right middle fingers), it entails the complex organization of motor commands. Consistent with non-human and human experiments demonstrating the necessity of task complexity for SMA activation[41,49], rTMS treatment over SMA did not influence

simple motor tasks, such as in-phase bimanual or unilateral finger tapping, whereas it inhibited the improvement in anti-phase bimanual tapping stability. By contrast, in the sham group, participants exhibited improvement in anti-phase bimanual tapping stability as well as a decrease in pre-movement M1-M1 PLV in the post-evaluation session. Moreover, multiple regression models of individual performance variability indicated the statistical significance of the interaction of SMA ERSP and M1-M1 PLV in the sham group, but not in the treatment group.

One potential explanation for the loss of interaction effects on anti-phase stability is that rTMS treatment might have prevented the influence of SMA on bi-hemispheric M1 phase synchrony, which decreased after offline consolidation in the sham group (Fig. 4b). According to the fitted model of the sham-stimulation group, lower ERSP magnitude of SMA and decreased pre-movement phase synchrony of bilateral M1, which occurred as a result of adaptation, were beneficial for subsequent performance[45]. This is probably because, to improve bimanual coordination performance, offline consolidation of the interaction of SMA dynamics with M1 is necessary for bi-hemispheric M1 to prevent their interference and process distinct motor commands sent from higher motor areas (Fig. 5a). On the other hand, in the treatment group, the interaction of ERSP magnitude at SMA and M1-M1 PLV did not influence anti-phase bimanual coordination because M1-M1 synchrony could not be led to the lower state without the modulatory effects of SMA (Fig. 5b).

Given that the influence of SMA on anti-phase coordination was mediated by M1-M1 phase synchrony, the results of experiment 1 that M1-M1 PLV-dependent movement initiation

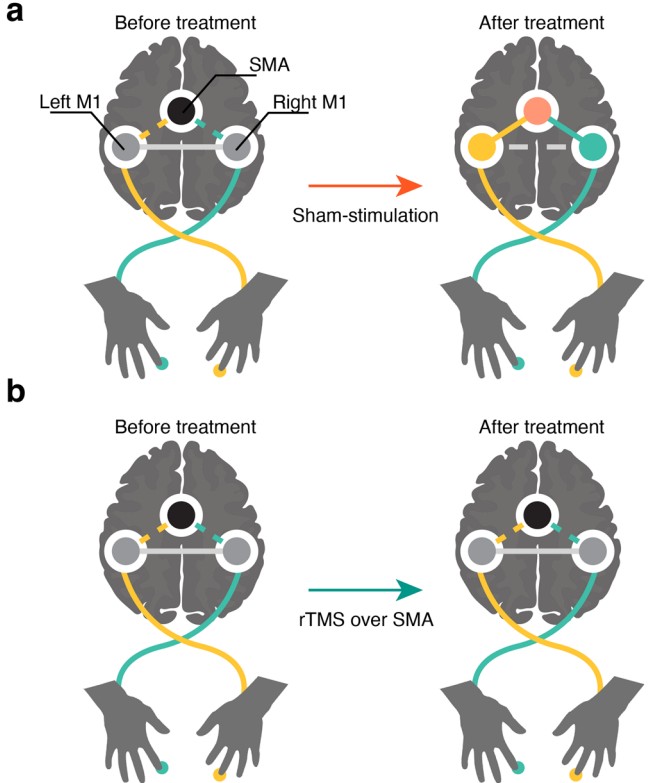

**Fig. 5 Schematic of interpretations of experiment results. a** Changes in activity patterns of the motor network by performance improvement. The interaction term of SMA activities and M1-M1 PLV reflect behavioral performance since the modulatory effects of SMA on M1 are weaker when M1-M1 PLV is low during the pre-movement period. **b** Changes in the activity patterns of the motor network after rTMS treatment. The interaction term of SMA activities and M1-M1 PLV do not reflect the performance since SMA activities do not modulate M1-M1 PLV due to SMA activity suppression.

modulated subsequent motor performance may be explained by an identical mechanism; lower M1-M1 phase synchrony prevented bi-hemispheric interference of motor commands from upstream of the cortical motor network and vice versa. Since the phase synchrony of distant brain regions underlies selective neural communication[25,26,50–52], lower M1-M1 PLV would be beneficial for the independent activation of bilateral fingers without the interference of motor commands. It should be noted that even though a systematic decrease in bi-hemispheric M1 phase synchrony was found after offline consolidation in the sham group in experiment 2, performance was related to the M1-M1 phase synchrony, as indicated by the significant main effect of M1-M1 PLV in the multiple regression model. This posits that these fluctuations of phase synchrony are incapable of fully control within a short-term period and influences bimanual coordination performance putatively due to individual difference in structural and synaptic properties of the brain[53–55].

Using SMR components as oscillatory scaffolds, the sensorimotor system entails large-scale neural communication[9,11,24,37,51,56–64]. Although SMR components found in alpha- and beta-bands share certain characteristics, such as event-related attenuation of signal power[65], their difference in functional relevance to the internal processing and cortical sources has recently been identified[66–69]. For instance, SMR in the alpha-band reflects sensory processing, such as gating of afferent information[68], while that in the beta-band reflects the efficacy of corticomotor output[9,61,70,71]. Based on previous reports that pre-movement modulation of alpha-

band activities mainly contributes to inhibiting undesired actions rather than the prepared action itself[68], the functional difference in the two frequency bands would account for beta-band dominant relationships between the pre-movement SMR dynamics and subsequent behavior. Since the beta-band activities were preferentially found in the cortico-cortical and cortico-subcortical motor networks, which shape output-related aspects of human behavior such as movement initiation or cancellation[10,11], the beta-dominant findings suggest that cortical pre-movement interaction organizes motor network to support subsequent stability of bimanual coordination[29]. Specifically, the pre-movement activities which lead bilateral M1 to the action-specific state would determine the initial state of the motor network and interference of bilateral corticomotor representation, which emerges through consolidation of acquired motor program[49,72]. Since the beta-band oscillations derived from scalp EEG signals represent motor cortical activities that regulate corticomotor output[73–75], the cortico-cortical communication tested with the M1-M1 phase coupling in the beta-band at the pre-movement period would reflect the initial state of motor network, which influences subsequent motor performance (duration of finger tapping) putatively not by directly modulating the transition-related activity separated in time, but by attenuating the interference of prepared motor program ideally distinctly represented in bilateral M1 to sustain the simultaneous activation of left index and right middle finger muscles and vice versa[24]. Indeed, the effects of interaction between the M1-M1 PLV and SMA on the duration of anti-phase bimanual tapping were dominantly observed after short-term break. It should be because the acquisition of motor program, which is the modulatory activities from higher-motor cortices[40,41,47], requires consolidation to make activities during preparatory period possible to influence the subsequent performance. Moreover, the result that rTMS treatment group did not exhibit the performance improvement suggests the acquired preparatory activity patterns (lower M1-M1 PLV and SMA spectral power) is necessary to achieve sustained anti-phase bimanual tapping.

Recent advances in the real-time processing of neural activities offer opportunities to improve behavior through its manipulation[7,9,20]. For instance, neurofeedback training is a task in which participants try self-regulation of intrinsic neural activities and bring it into a behaviorally beneficial state[76]. It has been demonstrated that SMR modulation can enhance behavioral performance, such as simple reaction time[9,71] and bimanual motor performance[20]. In our study, no feedback procedure was implemented to actively manipulate intrinsic activities, and the brain state-dependent movement initiations modulated the performance of bimanual coordination. The findings from the present study have potential application to human skill augmentation based on the mechanistic understanding of the motor network. However, studies on the behavioral effects of interaction between implicitly fluctuating and explicitly controlled neural activities are warranted[77,78].

In the present study, we used several non-invasive approaches to map neural activities and behavioral performance. Although cross-experiment replications suggested that our results are robust (Fig. 3b, Supplementary Fig. 4, Fig. 3c, and 4e), there were methodological limitations in both experiments. In experiment 1, the online estimation of M1-M1 PLV was conducted in sensor-space, i.e., the spatial and temporal filtering methods were optimized for scalp EEG data from the first session (See " Filtering parameter optimization" in Methods). Although the Hjorth montage extracts local neural oscillations under the electrode[59,79,80], the possibility that neural oscillations outside M1 or zero-lag phase synchrony due to volume conduction of EEG influence online PLV estimation cannot be excluded. Moreover,

although a significant difference in within-participant performance was demonstrated by M1-M1 PLV-based movement cueing, we did not test the relationship between performance and pre-movement SMA-ERSP using a similar state-dependent paradigm. Because it is not guaranteed that an inverse filter constructed for a single session is applicable to others, the real-time estimation of cortical source activities may not be readily achieved by scalp EEG recording. In experiment 2, the stimulation site during rTMS treatment was continuously monitored by experimenters using a neuronavigation system. However, we did not verify whether SMA was particularly stimulated using non-invasive brain stimulation. Although the evaluation of rTMS treatment using modulation of muscle-evoked potentials (MEPs) could inform the region-specific modulatory effect in M1[81,82], it would be challenging to apply a similar approach to SMA, which does not elicit muscle-specific MEP responses. Given that a number of studies using similar coil configurations reported successful manipulation of SMA activities, as manifested by behavioral changes distinct from those of M1[15,22,42], the stimulation protocol used in the present study likely targeted SMA activities. However, future confirmatory studies should apply more precise stimulation techniques, such as intracortical stimulation.

In summary, we showed that the manipulation of movement onset based on pre-movement intrinsic brain rhythmicity improved the ability to maintain anti-phase bimanual coordination after consolidation. Moreover, external perturbation over SMA inhibited the reorganization of pre-movement cortical interaction as well as behavioral improvement. This dynamic process ensures that oscillatory cortical activities influence bimanual motor performance without the subjects' awareness of the variation, suggesting the feasibility of augmenting human motor ability by covert control of movement cues based on the macroscopic neural coupling.

## Methods

**Participants**. A total of 62 participants without a history of neurological or psychiatric diseases (all right-handed according to the Edinburgh handedness inventory;[83] laterality quotient: 84.5 ± 21) were recruited. No participant reported more than one year of experience in piano playing. Two cohorts of participants, consisting of 30 (7 females and 23 males; 23.7 ± 2.9 years; age range: 20–35 years) and 32 (14 females and 18 males; 25.2 ± 7.9 years; age range: 20–57 years) participants performed experiments 1 and 2, respectively.

The sample size of experiment 1 was determined based on previous neuroimaging studies using a similar experimental paradigm aiming to test the within-participant effect of intervention[9,13,15,29]. Since that large effect size (at least Cohen's $d = 1.0$) was consistently reported, we calculated that 13 participants were needed based on a priori power analysis using G*Power 3.1[84] (α = 0.05, 1-β = 0.8, Bonferroni corrected). Finally, we recruited 15 participants for each group by considering exclusion due to the insufficient number of trials for statistical analyses (assumed 20% dropout rate[57,85]). That of experiment 2 was calculated from an a priori power analysis based on the effect size of changes in behavioral performance observed in experiment 1 (Cohen's $d = 0.78$, Fig. 1d); we calculated that 15 participants were needed for each group.

The study protocol was approved by the local Ethics Committee of the Faculty of Science and Technology of Keio University (IRB approval number: 2021-133). The experiments conformed to the Declaration of Helsinki and were performed in accordance with infection control protocols for EEG research[86]. All subjects provided written informed consent and reported their physical condition before participation.

**Experimental system**. The behavioral tasks in experiments 1 and 2 were conducted using the same experimental setup and custom scripts for MATLAB 2020a (The Mathworks, Inc, MA, USA). The behavioral data (i.e., information on keyboard tapping) were collected using Psychtoolbox extensions[87].

EEG and surface electromyogram (EMG) data were collected at a sampling rate of 1 kHz with a biosignal amplifier (GES 400 and Physio16; EGI, Eugene, OR, USA). Scalp EEG signals were recorded from a 128-channel wet electrode cap (HydroCel Geodesic Sensor Net; HCGSN, EGI, Eugene, USA). The layout of channels followed the international 10-10 electrode positions with the Cz channel set as the reference channel. The impedance of channels was maintained below 50 kΩ throughout the experiment. During EEG recording, EMG signals from bilateral

flexor digitorum superficialis muscles were concurrently recorded using Ag/AgCl electrodes to monitor finger movements during the ready period.

In addition to the EEG setup, a TMS experimental setup was employed in experiment 2. The setup comprised a TMS stimulator (MagPro X100 with magoptions, MagVenture, Alpharetta, GA, USA), neuronavigation system (Brainsight, Rogue Research, Montreal, Canada), and EMG recorder (Neuropack X1, Nihon Kohden, Tokyo, Japan). An active cooling butterfly coil with electrical skin stimulation (Cool-B65 A/P CO, MagVenture, Alpharetta, GA, USA) was used as the TMS coil to achieve sham and treatment (veritable) stimulation in a single-blinded manner.

**Experimental protocol**. Experiment 1 was conducted using a real-time, brain-state-dependent EEG-triggered movement cue system. This EEG-triggered setup leveraged analysis on online filtered EEG signals to trigger movement cues depending on the PLV of the recorded SMR signals from bilateral SM1, which represents inter-hemispheric phase synchrony of neural oscillations[31,59].

Participants underwent six sessions of ani-phase bimanual tapping task with EEG measurement (Fig. 1a). The first session was a state-independent condition in which task periods started independently from EEG. The remaining five sessions were state-dependent conditions using online EEG signal processing (see "Online signal processing"). Each session consisted of a 15-s resting state and 16 trials of anti-phase bimanual tapping task with EEG recording. In both conditions, a single trial comprised three periods: a 5-s rest period, a ready period with variable duration between trials (randomly sampled from $N[1,1]$), and a 20-s task period. An inter-trial interval lasted for 3 s after every task period.

During the task period, participants were instructed to perform anti-phase bimanual tapping with the timing of the tapping matched with the sound (one of the pairs was tapped at the presentation of one sound). As the phase transition of anti-phase bimanual movement occurs in a frequency-dependent manner[28,30], the tapping speed, which robustly induced phase transition within the task duration, was sought according to the adaptive method. Therefore, participants performed the tapping task paced by a 2-Hz beep sound for the first trial. In the next trial, the frequency was increased by +0.5 Hz if the duration of anti-phase exceeded 90% of the task period. Participants were alerted at the end of the trial in case the performance did not meet the following criteria: mean speed of finger tapping was more than 90% of the presented sound or the duration of anti-phase tapping lasted for at least 2 s.

In experiment 2, participants underwent four experimental blocks: a TMS block for threshold determination, a pre-evaluation block of motor performance with EEG acquisition, an rTMS block, and a post-evaluation block identical to the pre-evaluation session. First, participants underwent threshold determination for the subsequent rTMS block. The resting motor threshold (RMT) was determined for the right first dorsal interosseous muscle at its hot spot over the left M1. The RMT was determined as the stimulation intensity that elicited muscle-evoked potentials of greater than 50 μV during 5 out of 10 consecutive stimulations. The TMS block was followed by the pre-evaluation block. Participants first underwent habituation during the anti-phase bimanual tapping. To introduce the task, participants performed 5 trials of anti-phase bimanual tapping paced by a 2-Hz beep sound. The evaluation block consisted of four sessions: adaptive, fixed, in-phase, and unilateral finger-tapping sessions.

During the adaptive and fixed sessions, participants were asked to perform the anti-phase bimanual tapping task. The time-course of the trial was identical to that of experiment 1. First, the participants underwent the adaptive session in which they performed the anti-phase bimanual tapping at a variable speed. The first trial began with a 2-Hz sound, followed by faster sounds (+0.5 Hz) in subsequent trials if the participants maintained anti-phase coordination for more than 90% of a task period. The session was terminated if the participants failed to fulfil the anti-phase criteria for two consecutive trials. In the fixed session, the speed of the beep sound was fixed to test the anti-phase bimanual coordination performance at participant-wise-adjusted speed; participants performed 12 trials of anti-phase bimanual tapping at the final frequency used in the adaptive sessions.

In the subsequent in-phase and unilateral sessions, participants were instructed to perform in-phase bimanual and unilateral finger-tapping tasks, respectively. The time courses of a trial in the two sessions were identical to that during the anti-phase session. Participants performed 6 trials of in-phase finger tapping (i.e., tapping bilateral index and middle fingers in turn). In the unilateral session, visual instructions were provided to perform right or left-finger tapping during the ready period. Participants performed the task at a frequency identical to that during fixed and in-phase sessions. Participants underwent 6 trials for each side, and the order of trials for the right and left hand was pseudo-randomized. The evaluation procedure was performed before and after rTMS treatment.

In the rTMS block, we used 1-Hz rTMS to induce transient neural perturbation over SMA. The TMS coil was applied to the MNI coordinate [−4, -2, 56], which is the peak voxel determined by automated meta-analysis using Neuroquery[88]. The search word "SMA proper" was used to determine the stimulation target, and the papers included in the meta-analysis are listed in Supplementary Table 1. Participants underwent a 25-min 1-Hz rTMS session at 90% RMT using treatment or placebo side of the TMS coil with stimulation-locked forehead skin electrical stimulation. Participants were blinded for the allocated stimulation type and the existence of sham stimulation was not informed. In total, 16 participants were

allocated to each condition. During the rTMS session, the distance between the coil and the targeted site over the participant's head was displayed to the participants using a neuronavigation system. Participants were instructed to minimize head movement during stimulation and correct it when the distance increased (Average error, Treatment: $1.50 \pm 0.5$, Sham: $1.75 \pm 0.7$ mm). After the rTMS session, participants reported sleepiness using the Stanford Sleepiness Scale[89] (treatment: $4.00 \pm 1.7$, sham: $4.71 \pm 1.4$).

After the rTMS treatment, participants wore the EEG caps and underwent a post-evaluation block identical to the pre-evaluation block. However, in the adaptive session of the post-evaluation block, the frequency of the presented sound was identical to that for the pre-treatment session, regardless of participants' performance, to control for potential effects on performance during the subsequent fixed session.

**Behavioral data analysis**. The behavioral data were preprocessed to identify the onset of each tap and the timing of phase transition. The timing of phase transition was based on the previous studies employing an identical task[13]. In brief, the difference in key press timing of each finger (hereinafter RI: right index, LI: left index, RM: right middle, LM: left middle) was compared with the corresponding pairs of the anti-phase (e.g., RI and LM) and in-phase (e.g., RI and LI) tapping to identify the point that satisfies the following condition:

$$\min\left(\left|RI(i) - LM\right|\right) > \min\left(\left|RI(i) - LI\right|\right) \tag{1}$$

where $RI(i)$, $i$-th tapping timing, $LM$, and $LI$ indicate vectors of tapping timing in a trial. The first phase transition is identified as $T_{RI} = RI(i_{trans})$, where $i_{trans}$ is the first $i$ that satisfies the inequality above. For each pair of anti-phase tapping, identical calculations were performed to derive $T_{LI}$, $T_{RM}$, $T_{LM}$. Finally, a maximum of four time points ($T_{trans}$) were adopted as the phase transition in a trial. In the case that none of the pairs satisfied the inequality, $T_{trans}$ was operationally set as 20, which was the total length of the task period. After the calculation of $T_{trans}$, the duration of anti-phase tapping was calculated as the difference between the timing of the first tapping in a trial $T_{init} = \min(RI(1), LI(1), RM(1), LM(1))$ and $T_{trans}$. To normalize the task difficulty across participants, trials that presented beep sound at a mode frequency for each participant were extracted. Data from participants containing at least 5 trials were included in statistical analysis.

In experiment 2, in addition to the duration of anti-phase bimanual tapping, the variability of inter-tap interval (ITI) was measured to assess the performance of rhythmic production of tapping tasks for other types of tasks. ITI was calculated for the data from each finger used in the task; in tasks using multiple fingers, the ITI was averaged to calculate a representative value for a single trial.

**EEG data analysis**

*Filtering parameter optimization.* In experiment 1, after the first session, the recorded EEG signals were subjected to a calibration process to determine the optimal filtering parameters for subsequent triggered sessions. EEG signals were filtered using a third-order Butterworth notch (50 Hz) and bandpass filtered (1–45 Hz). The epoched EEG signals were subjected to short-term Fourier transformation to calculate the band power in alpha (8–13 Hz) or beta (14–30 Hz) bands, which were averaged across trials. The responsive frequency range was determined by the magnitude of ERSP from one of the two frequency bands (i.e., alpha or beta), which was pseudo-randomly allocated to each participant (15 participants for each frequency band). For each frequency band (alpha and beta), the averaged magnitude of ERSP was calculated in 8–13 Hz for the alpha and 14–30 Hz for the beta at 3 Hz sliding bin and 2 Hz overlap. The frequency of interest (FOI) was determined by that which exhibited minima of ERSP magnitude in C3 and C4 channels (averaged across time at task period and channels) since the power attenuation of SMR around bilateral M1 reflects corticospinal excitability[90–92]. Then, the topographic representation of the frequency-of-interest (FOI) was visualized. The channel-by-channel median values of ERSP magnitude were compared within the six candidate channels around C3 and C4, as well as amongst themselves to determine the channels-of-interest (COIs) from bilateral temporal regions. After determining the COIs from bilateral hemispheres, the parameters for online filtering and forward prediction with autoregressive (AR) models were determined based on genetic algorithms[58]. During calibration, participants were asked to rest for around 15 min without practicing the anti-phase bimanual tapping.

*Online signal processing.* In experiment 1, we used online PLV-dependent movement cueing to inform participants to begin the anti-phase bimanual tapping tasks in a brain-state dependent manner. In the state-dependent sessions, the measured EEG signals were subjected to custom MATLAB scripts and processed in the following manner; the bandpass filtered EEG signals around bilateral SM1 were processed with a Hjorth-style Laplacian spatial filter, and the filter calibrated for FOI of each participant. Then, online calculated PLV was used to monitor phase synchronicity between bilateral SM1 in the targeted frequency bands (i.e., FOI of alpha or beta bands).

$$PLV = abs\left(\frac{1}{N}\sum_{t=1}^{N} \exp\left(i\left(\phi_{Lefthemisphere} - \phi_{Righthemisphere}\right)\right)\right) \tag{2}$$

where $N$ is the size of a time window for PLV calculation, $i$ is the imaginary unit, and $\phi$ is the instantaneous phase of targeted regions.

During the ready period, if the PLV values satisfied the threshold for triggering high and low conditions, the task period began immediately. To determine the thresholds at the beginning of each session, participants underwent a 15-s rest. After outlier detection using median absolute deviation for all data samples (MAD, points deviating more than 3 times the MAD were rejected), 25% and 75% quartiles of PLV distribution were set as the threshold for low and high PLV states, respectively. The PLVs were calculated after every 100 ms using the latest 1-s EEG signals and averaged with the previous 10 points, and intermittent beep sound was produced when the PLV exceeded the threshold for the high condition or dropped below that of low. In case the moving-averaged PLVs did not reach the upper and lower thresholds during the ready period, the trial was performed identically to the state-independent conditions.

*Offline analysis.* For offline analysis, the measured EEG signals were subjected to the following analytic pipeline to estimate cortical source activities. First, EEG signals were preprocessed with band pass and notch filters, similar to online analysis. Then, using EEGLAB[93] and its plugins, independent component analysis was performed. Based on the automatic labeling algorithm, components identified as non-neural activity were subtracted[94]. Then, the EEG signals processed with common average reference were subjected to cortical source estimation using Brainstorm toolbox[95] with the normalized brain and sLoreta algorithm[96]. After the reconstruction of source activities, the time series of mean activities in SMA and bilateral precentral gyri were extracted. Then, the time-frequency analysis with wavelet transformation was applied to extract the band power in alpha and beta bands. Using the extracted band power, ERSP magnitude was calculated using the rest period as reference to test the difference in the responsive region between two bands[65].

$$ERSP(f,t), \% = 100 \times \frac{\left(A(f,t) - Ref(f)\right)}{Ref(f)} \tag{3}$$

where $A(f,t)$ is the signal strength at a time point $t$ in the frequency $f$ and $Ref(f)$ is the reference signal strength in the frequency $f$. $Ref(f)$ was calculated as the averaged power during the rest period. PLV between the estimated cortical activities in bilateral M1 were calculated using a procedure identical to that of online analysis and subjected to offline analysis. In the control analysis, alternative lengths of time-window and phase coupling metrics were employed to test the robustness of the results. In the former analysis, a longer window (1200 ms) and a shorter window (800 ms) were applied to the source-space signals. In the latter analysis, imaginary-PLV (iPLV) was employed to test the effects of the zero-lag connectivity which induces spurious phase-coupling due to volume conduction of EEG signals, since iPLV values are insensitive to the zero-lag connectivity by eliminating real-part from the PLV metrics as follows:

$$iPLV = abs\left(\frac{1}{N}\sum_{t=1}^{N} imag\left(\exp\left(i\left(\phi_{Lefthemisphere} - \phi_{Righthemisphere}\right)\right)\right)\right) \tag{4}$$

where $N$ is the size of a time window for PLV calculation, $i$ is the imaginary unit, $\phi$ is the instantaneous phase of targeted regions and the imag operator extracts the imaginary part of the phase difference[97].

**Statistical analysis**. Statistical tests were conducted using JASP[98], R (ver. 4.0.2), and MATLAB (ver. 2021b). Shapiro-Wilk test was performed to test for data normality. If the null hypothesis was rejected ($p < 0.05$), corresponding non-parametric tests were applied. In experiment 1, the difference in anti-phase duration between state-independent sessions was compared to that in the first state-dependent session. The duration across state-dependent sessions was subjected to one-way ANOVA. To validate whether the online PLV-dependent trigger effectively differed between the two conditions (i.e., PLV high and low conditions), pre-movement M1-M1 PLV and duration were averaged within participants and compared using a paired $t$-test. The relationship between the duration of anti-phase bimanual tapping and instantaneous phase synchrony of bilateral SMR was tested using Pearson's correlation analysis. The comparison was performed for participants whose data were obtained from at least 5 trials in both PLV high and low conditions. The inclusion criterion was determined based on results of a preliminary experiment to quantify the trial-to-trial variability of anti-phase tapping performance (Supplementary Fig. 11). The between-participant association of M1-M1 PLV and anti-phase tapping duration were tested using a repeated-measure correlation analysis for data obtained under high and low conditions[36]. The ERSP magnitude at SMA in alpha- and beta-bands were subjected to a mixed two-way repeated measures ANOVA with "Condition" and "Group" as the main effects. The between-participant association of ERSP magnitude at SMA and anti-phase tapping duration were tested using a Pearson's correlation analysis. For each participant, averaged data were calculated from trials in state-dependent conditions (i.e., sessions 2–5), which did not begin based on M1-M1 PLV. Using the same dataset, multiple regression analysis was performed in a group-by-group manner using ERSP magnitude at SMA and M1-M1 PLV as explanatory variables, and duration of anti-phase bimanual tapping as response variables.

In experiment 2, the behavioral data (duration of anti-phase bimanual tapping and ITI variability) and pre-movement EEG data (ERSP magnitude and M1-M1

PLV) were subjected to a mixed two-way repeated measures ANOVA with "Time" and "Group" as the main effects. In the test for M1-M1 PLV, the baseline ERSP magnitude at SMA and changes in duration were considered as covariates to account for the variability in stimulation effects. If statistical significance was observed, a post hoc t-test with Bonferroni correction was applied. For the confirmation of baseline homogeneity, data from pre-evaluation sessions were compared between groups using a two-sample *t*-test. The between-participant association of M1-M1 PLV, pre-movement SMR-ERD magnitude from SMA, and anti-phase tapping duration was tested, similar to that for experiment 1. Similarly, the relationship between changes in the duration and ERSP magnitude at SMA was tested using Spearman's correlation analysis. For multiple regression analysis using ERSP magnitude at SMA and M1-M1 PLV as explanatory variables, mixed-effect models with normal distribution as the link function for post-evaluation data to consider the tapping frequency as a random effect[99].

**Reporting summary**. Further information on research design is available in the Nature Portfolio Reporting Summary linked to this article.

## Data availability

All data supporting the findings of the study are available from figshare (https://figshare.com/projects/Beta_rhythmicity_in_human_motor_cortex_reflects_neural_population_coupling_that_modulates_subsequent_finger_coordination_stability/144057).

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

## Acknowledgements

The present study was supported by JST CREST (JPMJCR17A3) including the AIP challenge program and JSPS KAKENHI Grant Number 21J20955. We thank Shoko Tonomoto, Aya Kamiya, and Yui Yoshioka for their general support.

## Author contributions

Conceptualization: S.I., T.Y., and J.U.; methodology: S.I., T.Y., and J.U.; investigation: S.I., T.Y., R.H., and J.U.; data curation, formal analysis, and software: S.I. and R.H.; funding acquisition: S.I., T.Y., and J.U.; writing–original draft: S.I. and T.Y.; writing–review and editing: S.I., T.Y., R.H., and J.U.; supervision: J.U.

## Competing interests

J.U. is the founder and representative director of the university startup company, LIFESCAPES Inc., which is involved in the research, development, and sales of rehabilitation devices, including brain-computer interfaces. He receives a salary from LIFESCAPES Inc., and holds shares in LIFESCAPES Inc. This company does not have any relationships with the device or setup used in the current study. The remaining authors declare no competing interests.
