## [Peer Review File · Communications Biology]

Reviewers' comments:

Reviewer #1 (Remarks to the Author):

Iwama and colleagues report two experiments in human subjects that address the role of cortico-cortical rhythms in the control of bi-manual movement. While previous work has implicated increases in M1-M1 functional connectivity prior to movement onset in bi-manual coordination, and further suggests a pivotal role of the SMA in this behavior, interventional evidence has been largely lacking. Here the authors address this. First, they use EEG to synchronize behavior with epochs of high and low M1-M1 functional connectivity, respectively. If such pre-movement connectivity was indeed relevant for bi-manual coordination, then differences in behavior ought to be observed between these conditions, specifically in the alpha or beta bands, which were both assessed. Second, if SMA is indeed an additional crucial node in bi-manual coordination, then transiently perturbing its function (here using TMS) ought to impair the behavioural benefits seen with prolonged behavior. Both predictions were confirmed in separate, independent sets of experiment. These findings suggest that inter-regional changes in functional connectivity prior to movement onset play a crucial role in coordinating bimanual movement in human subjects, with specific relevance of the SMA.

My main comments pertain to the EEG analyses. The authors use the phase-locking value (PLV) as an index of functional connectivity. This measure is not without limitations and further reassurance that the results are not contaminated by some of these issues seems warranted, especially as much of the results relies on this measure.

1. The results of PLV analyses can be strongly influenced by the length and period of the signal used. It would be reassuring to see this addressed, ie to know that the results are not falling apart when these parameters are changed.
2. A general concern with PLV measures is volume conduction, ie a single source can be seen by both the electrodes resulting in spurious PLV. This is here of concern because the two M1 regions and the SMA which are used for these analyses are relatively close (in EEG terms). A formal comparison of the PLV measures that were obtained online and offline would be useful to reassure the reader that the online triggering of behavior based on PLV was indeed valid. One concern is zero-lag connectivity, and the degree to which this may have contributed would be reassuring to know. One example is that the PLV values for M1-M1 alpha seem rather large - this may be indeed what is happening, ie large phase synchrony between regions, but control analyses reassuring this is the case would help to alleviate any concerns on this issue.
3. The use of a common reference at Cz might be additionally problematic, and can pose a known problem for PLV analyses.
4. For their offline analyses, the authors assess activity in M1 and SMA in source space. This does not mean that volume conduction can be ruled out however. However, the paper is relatively sparse on detail for this analysis and the results. It would be reassuring to see the source data in a lot more detail. Put simply, the proximity of the three brain regions of interest make signal leakage a real concern. At times it is also not clear whether the reported results are all based on the offline source data, or whether some results are derived from sensor space analyses.
5. One way to address these concerns is to employ surrogate data and analyses, and one would hope to see these for both alpha and beta.

The paper is somewhat vague on the biological mechanism at hand. In essence, it relies on a rather vague notion of pre-movement 'interactions'. Some questions come to mind: what is the specific role of beta (vs other frequencies) in pre-movement activity? The suggestions that "beta-dominant findings suggest that cortical pre-movement interaction organizes motor network to support subsequent stability of bimanual coordination" is rather vague and hard to falsify. How would, for example, a change in beta in this network before the onset of bi-manual coordinated movement – lasting 10s of seconds as here – provide a behavioural advantage? There is an obvious reason for focusing on pre-movement activity, as this is uncontaminated by overt movement, but it raises the question how that activity influences an nth movement occurring a relatively long time later? More broadly, what is happening physiologically prior to movement that leads to (if true) changes in bimanual behavior? Related, what are the physiological mechanisms that lead to PLV increases and how would that potentially contribute to later performance changes? I appreciate the answers may not be known; a slightly more concrete interpretation of how a signal in a network at one point (pre movement) can lead to the behavioural changes seen 10s of

seconds later would seem relevant.

Other comments:

I found the paper overall accessible and nice to read but encourage additional proof reading for minor grammatical errors.

On line 273, it seems that an interaction in the 2-way ANOVA was followed up with post-hoc ttests. To the best of my knowledge, this is not the approach for dealing with interactions. Please check your analyses throughout.

The data sharing statement seems rather weak and untimely. For a publication of this kind one might expect a lot more transparency and openness.

Reviewer #2 (Remarks to the Author):

I really want to find important research significance from this paper. But I find it is difficult. Don't blame me. The paper describes a series of experiments that are not very meaningful in a very difficult way. Only two related frequencies are analyzed. It does not explain why only these two frequencies are important to the experiment. The calculation method is too concise and seems to include only PLV and variance statistics.

To be fair, maybe we need other reviewers.

Please accept my apology.

Reviewer #3 (Remarks to the Author):

This is the study investigating whether pre-movement cortical interaction in the motor network modulates subsequent bimanual coordination performance. The authors reported the negative correlation between the duration of anti-phase bimanual tapping and pre-movement M1-M1 PLV, and the significant effects of M1-M1 PLV and ERSP at the SMA on the duration. Further, they reported that the effects of the interaction and performance improvement were disrupted by low-frequency rTMS over the SMA.

This study provides an important contribution to the understanding of the effects of the dynamics of pre-movement neural oscillations in the motor network on bimanual coordination. However, a revision is needed before it can be accepted for publication.

Major

This study investigated anti-phase bimanual coordination stability after consolidation in Experiment 1 and perturbed the consolidation in Experiment 2. The findings about the correlation among the duration of anti-phase bimanual tapping, pre-movement M1-M1 PLV, and ERSP at the SMA were limited to the state after the consolidation. Actually the authors did not find the significant correlation among them in Pre-stimulation in Experiment 2 (Supplementary Fig. 4). When they described the findings in Abstract and Discussion, they should refer to the limitation. I think correlation analysis and multiple regression analysis in this study should be performed separately for the data before the consolidation and those after the consolidation.

I found the descriptions such as 'manipulation of pre-movement M1-M1 PLV' in the manuscript. However, the authors did not manipulate the PLV, but they manipulated the timing of the movement cues depending on the PLV. They should revise them.

Some participants showed higher M1-M1 PLV in PLV Low condition than in PLV High condition (Fig. 2a). Were these participants included in further analysis? I think they should be excluded from further analysis.

Minor

Fig.1

c Top: What does the yellow bar represent?

e: The authors should describe what the black lines represent (the same in other figures).

The black lines were missing in Session 4 and 5.

Page 8 Line 161

The authors should explain the rationale for using 5 trials as the inclusion criteria.

Fig.3

Are the data in Fig. 3 those from Session 2 – 5? I think they should not include those from Session 1. (Please see above)

Fig. 3 c

What does the blue curved surface represent?

Fig. 4

There is no description about which is pre- or post-evaluation (the same in other figures).

Fig. 4d and e

Are the data in Fig. 4d and e those from post-evaluation? I think they should not include those from pre-evaluation. (Please see above)

Fig. 4e

What do the green and orange curved surfaces represent?

Page 30

The authors should add description about how the responsive frequency and the COIs were determined. Was the responsive frequency the one showing maximum ERSP? Were the COI the one showing maximum median values of ERSP?

Page 30

Did the authors use all the EEG data (15 min) for calibration?

Supplementary Fig. 1f

Please explain why the lines representing PLV Low condition don't have a negative peak at (or around) Time 0.

Supplementary Fig. 7

I think the data in Supplementary Fig. 7 should not include those from pre-evaluation. (Please see above)

Typo

Page 14 Line 285

0.85. -> 0.85).

Page 15 Line 302

group Fig. 4d -> group (Fig. 4d

Point-by-point responses to comments from referees

We thank the referees and the editor for their support and constructive suggestions. We addressed all concerns of the referees in our updated manuscript. Please find detailed explanations below.

Referee #1:

Iwama and colleagues report two experiments in human subjects that address the role of cortico-cortical rhythms in the control of bi-manual movement. While previous work has implicated increases in M1-M1 functional connectivity prior to movement onset in bi-manual coordination, and further suggests a pivotal role of the SMA in this behavior, interventional evidence has been largely lacking. Here the authors address this. First, they use EEG to synchronize behavior with epochs of high and low M1-M1 functional connectivity, respectively. If such pre-movement connectivity was indeed relevant for bi-manual coordination, then differences in behavior ought to be observed between these conditions, specifically in the alpha or beta bands, which were both assessed. Second, if SMA is indeed an additional crucial node in bi-manual coordination, then transiently perturbing its function (here using TMS) ought to impair the behavioural benefits seen with prolonged behavior. Both predictions were confirmed in separate, independent sets of experiment. These findings suggest that inter-regional changes in functional connectivity prior to movement onset play a crucial role in coordinating bimanual movement in human subjects, with specific relevance of the SMA.

My main comments pertain to the EEG analyses. The authors use the phase-locking value (PLV) as an index of functional connectivity. This measure is not without limitations and further reassurance that the results are not contaminated by some of these issues seems warranted, especially as much of the results relies on this measure.

We thank for valuable and constructive comments for our manuscript. We employed additional analysis and revised our manuscript according to comments. Please find the point-by-point responses to your comments.

Major comments:

Comment 1.1

1. The results of PLV analyses can be strongly influenced by the length and period of the signal used. It would be reassuring to see this addressed, ie to know that the results are not falling apart when these parameters are changed.

In the original manuscript, we used the 1-s window for PLV calculation throughout two experiments and observed the correlation between pre-movement M1-M1 PLV and anti-phase tapping duration. We found

similar results when we used shorter (800 ms) and longer (1200 ms) windows, respectively.

Experiment1.

Fig. R1. Phase-locking values computed with an alternative time-window in Experiment 1.

Because the analysis is the supplement of Fig. 2a in the original manuscript, the identical statistical comparisons were conducted for data from each condition. Results of statistical tests corresponding figures are as follows:

Longer window (1200 ms):

Beta-group, Paired t-test, $t(14) = 5.70, p < 0.001, d = 1.47, CI_{95} = [0.7 \ 2.20]$

Alpha-group, Paired t-test, $t(14) = 2.63, p = 0.02, d = 0.68, CI_{95} = [0.1 \ 1.23]$

Shorter window (800 ms):

Beta-group, Paired t-test, $t(14) = 6.36, p < 0.001, d = 1.63, CI_{95} = [0.84 \ 2.42]$

Alpha-group, Paired t-test, $t(14) = 6.28, p < 0.001, d = 1.62, CI_{95} = [0.83 \ 2.39]$

Moreover, we analyzed the PLV-Duration correlation analysis using the PLV computed using the altered parameters. The results were preserved in both longer- and shorter-window in the beta-band (Fig. R2).

COMMSBIO-22-2454R1: Beta rhythmicity in human motor cortex reflects neural population coupling that modulates subsequent finger coordination stability

Fig. R2. Repeated measures correlation test using the alternative time-window parameters.

Because the analysis is the supplement of Fig. 2d in the original manuscript, the identical statistical comparisons were conducted for data from each condition. Results of statistical tests corresponding figures are as follows:

Longer window (1200 ms):

Beta-group, Repeated measures correlation test, $p = 0.04$, $r = -0.57$, $CI_{95} = [-0.87 \ 0.03]$

Alpha-group, Repeated measures correlation test, $p = 0.62$, $r = -0.15$, $CI_{95} = [-0.69 \ 0.49]$

Shorter window (800 ms):

Beta-group, Repeated measures correlation test, $p = 0.03$, $r = -0.59$, $CI_{95} = [-0.88, 0.01]$

Alpha-group, Repeated measures correlation test, $p = 0.37$, $r = -0.27$, $CI_{95} = [-0.74 \ 0.39]$

We implemented these results in the revised supplementary information (SI) as follows:

COMMSBIO-22-2454R1: Beta rhythmicity in human motor cortex reflects neural population coupling that modulates subsequent finger coordination stability

Supplementary Fig. 3 Supplementary analysis using alternative time-windows in Experiment 1

[Line 37 in SI] **a** Comparison of phase-locking values (PLVs) of bilateral primary motor cortices (M1) between two conditions, calculated with a longer time-window (1200 ms). Groups based on M1-M1 PLVs in both beta- and alpha-bands exhibited significant differences in high and low conditions (paired t-test, *: $p < 0.05$, beta-group: $t(14) = 5.70$, $d = 1.47$, $CI_{95} = [0.7 \ 2.20]$; alpha-group: $t(14) = 2.63$, $p = 0.02$, $d = 0.68$, $CI_{95} = [0.1 \ 1.23]$). **b** Comparison of M1-M1 PLVs between two conditions, calculated with a shorter time-window (800 ms). Groups based on M1-M1 PLVs in both beta- and alpha-bands exhibited significant differences in high and low conditions (paired t-test, *: $p < 0.05$, beta-group: $t(14) = 6.36$, $d = 1.63$, $CI_{95} = [0.84 \ 2.42]$; alpha-group: $t(14) = 6.28$, $d = 1.62$, $CI_{95} = [0.83 \ 2.39]$). **c** Correlation analysis between M1-M1 PLV (longer-time window) and duration (repeated measures correlation test, beta: $r = -0.57$, $p = 0.04$; alpha: $r = -0.15$, $p = 0.62$). Solid and dotted lines indicate linear regression and a 95% confidence interval, respectively. **d** Correlation analysis between M1-M1 PLV (shorter-time window) and duration (repeated measures correlation test, beta: $r = -0.59$, $p = 0.03$; alpha: $r = -0.27$, $p = 0.37$). Solid and dotted lines indicate linear regression and a 95% confidence interval, respectively.

Experiment 2.

For supplementary analysis of Experiment 2 data, we employed the alternative time-windows to test the robustness of the result. We found similar results when we used shorter (800 ms) and longer (1200 ms) windows, respectively.

Fig. R3. Phase-locking values computed with an alternative time-window in Experiment 2.

Because the analysis is the supplement of Fig. 4b in the original manuscript, the identical statistical comparisons were conducted for data from each condition. Results of statistical tests corresponding figures are as follows:

Longer window (1200 ms):

Beta-band:

Mixed two-way repeated measures ANOVA (The identical analysis employed in the original manuscript, please see Comment 1.8):

Interaction of “Time” × “Group”: $F(1, 28) = 4.37, p = 0.045, \eta^2 = 0.074$

Main effect of “Time”: $F(1, 28) = 2.40, p = 0.13$

Main effect of “Group”: $F(1, 28) = 0.019, p = 0.89$

Post-hoc *t*-test with Bonferroni correction:

Group-by-group comparison:

COMMSBIO-22-2454R1: Beta rhythmicity in human motor cortex reflects neural population coupling that modulates subsequent finger coordination stability

Treatment: Paired t -test, $t(15) = 0.525$, $p = 0.61$

Sham: Paired t -test, $t(15) = 3.61$, $p = 0.003$, $d = 0.90$, $CI_{95} = [0.31, 1.49]$

Time-by-time comparison:

Pre-evaluation: Two-sample t -test, $t(30) = -1.41$, $p = 0.17$

Post-evaluation: Two-sample t -test, $t(30) = 1.78$, $p = 0.09$

Alpha-band:

Mixed two-way repeated measures ANOVA:

Interaction of "Time" × "Group": $F(1, 28) = 0.18$, $p = 0.67$

Main effect of "Time": $F(1, 28) = 1.36$, $p = 0.25$

Main effect of "Group": $F(1, 28) = 0.007$, $p = 0.98$

Shorter window (800 ms):

Beta-band:

Mixed two-way repeated measures ANOVA:

Interaction of "Time" × "Group": $F(1, 28) = 4.24$, $p = 0.049$, $\eta^2 = 0.06$

Main effect of "Time": $F(1, 28) = 0.11$, $p = 0.75$

Main effect of "Group": $F(1, 28) = 0.11$, $p = 0.74$

Post-hoc t -test with Bonferroni correction:

Group-by-group comparison:

Treatment: Paired t -test, $t(15) = -0.023$, $p = 0.98$

Sham: Paired t -test, $t(15) = 3.43$, $p = 0.004$, $d = 0.86$, $CI_{95} = [0.27, 1.42]$

Time-by-time comparison:

Pre-evaluation: Two-sample t -test, $t(30) = -1.49$, $p = 0.15$

Post-evaluation: Two-sample t -test, $t(30) = 1.12$, $p = 0.27$

Alpha-band:

Mixed two-way repeated measures ANOVA:

Interaction of "Time" × "Group": $F(1, 28) = 0.86$, $p = 0.36$

Main effect of "Time": $F(1, 28) = 0.19$, $p = 0.67$

Main effect of "Group": $F(1, 28) = 0.77$, $p = 0.39$

We implemented these results in the revised supplementary information as shown in below. To simplify the captions, we did not report the exact statistical values for those not significant such as results in the alpha-band. However, if the referee feels it should be implemented, we are happy to do.

[Line 71 in SI] **Supplementary Fig. 5 Supplementary analysis using alternative time-windows in Experiment**

2

a Pre-movement M1-M1 PLVs in the beta-band calculated with a longer time-window (1200 ms). Differences in phase shuffled data were subjected to a mixed two-way repeated measures ANOVA. A significant interaction of “Time” × “Group” was observed in the beta-band ($F(1, 28) = 4.37, p = 0.045, \eta^2 = 0.074$) while no main effects of “Time” or “Group” were found (“Time”: $F(1, 28) = 2.40, p = 0.13$, “Group”: $F(1, 28) = 0.019, p = 0.89$). Significant difference in the sham group revealed by post hoc *t*-tests (*: $p < 0.05, t(15) = 3.61, p = 0.003, d = 0.90, CI_{95} = [0.31, 1.49]$) For the identical analysis on the alpha-band data, no significant interaction and main effects were found (all $p > 0.05$). **b** Pre-movement M1-M1 PLVs in the beta-band calculated with a shorter time-window (800 ms). Differences in phase shuffled data were subjected to a mixed two-way repeated measures ANOVA. A significant interaction of “Time” × “Group” was observed in the beta-band ($F(1, 28) = 4.24, p = 0.049, \eta^2 = 0.06$) while no main effects of “Time” or “Group” were found (“Time”: $F(1, 28) = 0.11, p = 0.75$, “Group”: $F(1, 28) = 0.11, p = 0.74$). Significant difference in the sham group revealed by post hoc *t*-tests (*: $p < 0.05, t(15) = 3.43, p = 0.004, d = 0.86, CI_{95} = [0.27, 1.42]$) For the identical analysis on the alpha-band data, no significant interaction and main effects were found (all $p > 0.05$).

[Line 728] In the control analysis, alternative length of time-window and phase coupling metric were employed to test the robustness of the results. In the former analysis, a longer window (1200 ms) and a shorter window (800 ms) were applied to the source-space signals.

Comment 1.2

2. A general concern with PLV measures is volume conduction, ie a single source can be seen by both the electrodes resulting in spurious PLV. This is here of concern because the two M1 regions and the SMA which are used for these analyses are relatively close (in EEG terms). A formal comparison of the PLV measures that were obtained online and offline would be useful to reassure the reader that the online triggering of behavior based on PLV was indeed valid. One concern is zero-lag connectivity, and the degree to which this may have contributed would be reassuring to know. One example is that the PLV values for M1-M1 alpha seem rather large - this may be indeed what is happening, ie large phase synchrony between regions, but control analyses reassuring this is the case would help to alleviate any concerns on this issue.

According to the suggestion, we conducted the formal comparison of online-calculated, sensor-space M1-M1 PLV and offline-calculated, source-space M1-M1 PLV. As shown below, the similarity of two values derived from each participant in each condition (PLV-High and PLV-Low) were consistent.

COMMSBIO-22-2454R1: Beta rhythmicity in human motor cortex reflects neural population coupling that modulates subsequent finger coordination stability

Fig. R4. Comparison of M1-M1 PLV derived from sensor- and source-space signals.

Beta-group, Pearson’s correlation test, $p = 0.004$, $r = 0.50$, $CI_{95} = [0.17, 0.72]$

Alpha-group, Pearson’s correlation test, $p = 0.013$, $r = 0.45$, $CI_{95} = [0.10, 0.70]$

Moreover, to address the concerns about zero-lag connectivity, we used an alternative metric for the connectivity measure, corrected imaginary PLV (iPLV) (Bruña et al. 2018). By eliminating the real part from the calculation of phase synchrony, the zero-lag connectivity is effectively excluded. Even using the metrics, we found similar linear relationship with source-space PLV values derived by the original analysis pipeline. As the referee raised, the issue of zero-lag, spurious phase-locking due to signal leakage was of concern to measure “true” functional connectivity. However, we believe the zero-lag connectivity did not confound the present demonstration the alternative metric, iPLV yielded similar relationship between the behavioral performance as shown in Fig. R5.

Fig. R5. Comparison of M1-M1 PLV and M1-M1 iPLV.

a Beta-group, Pearson’s correlation test, $p < 0.001$, $r = 0.86$, $CI_{95} = [0.72, 0.93]$, Alpha-group, Pearson’s correlation test, $p < 0.001$, $r = 0.86$, $CI_{95} = [0.73, 0.93]$. **b** Beta-group, Repeated measures correlation test,

COMMSBIO-22-2454R1: Beta rhythmicity in human motor cortex reflects neural population coupling that modulates subsequent finger coordination stability

$p = 0.03$, $r = -0.59$, $CI_{95} = [-0.88, 0.02]$. Alpha-group, Repeated measures correlation test, $p = 0.16$, $r = -0.42$, $CI_{95} = [-0.81, 0.24]$

The supplementary analysis was implemented in the revised manuscript and supplementary information as follows:

Supplementary Fig. 4 Supplementary analysis using surrogate data and alternative metric of phase synchrony

[Line 55 in SI] **a** Comparison of M1-M1 PLV derived from sensor- and source-space signals. Pearson's correlation test was employed to test the consistency between the two datasets (Beta-group, $r = 0.50$, $p = 0.004$, $CI_{95} = [0.17, 0.72]$, Alpha-group, $r = 0.45$, $p = 0.013$, $CI_{95} = [0.10, 0.70]$). Solid and dotted lines indicate linear regression and 95% confidence interval, respectively. **b** Comparison of M1-M1 PLV and imaginary-PLV (iPLV) derived from source-space signals. Pearson's correlation test was employed to test the consistency between the two datasets (Beta-group, $r = 0.86$, $p < 0.001$, $CI_{95} = [0.72, 0.93]$; Alpha-group, $r = 0.86$, $p < 0.001$, $CI_{95} = [0.73, 0.93]$). **c** Correlation analysis between M1-M1 iPLV and duration of anti-phase tapping (repeated measures correlation test, beta: $r = -0.59$, $p = 0.03$; alpha: $r = -0.42$, $p = 0.16$).

[Line 731] In the latter analysis, imaginary-PLV (iPLV) was employed to test the effects of the zero-lag connectivity which induces spurious phase-coupling due to volume conduction of EEG signals, since iPLV values are insensitive to the zero-lag connectivity by eliminating real-part from the PLV metrics as follows:

$$iPLV = abs(1/N \sum_{t=1}^N imag(exp(i(\phi_{Left\ hemisphere} - \phi_{Right\ hemisphere}))))$$

where N is the size of time window for PLV calculation, i is the imaginary unit, ϕ is the instantaneous

COMMSBIO-22-2454R1: Beta rhythmicity in human motor cortex reflects neural population coupling that modulates subsequent finger coordination stability

phase of targeted regions and imag operator extracts the imaginary part of the phase difference⁹⁵.

Comment 1.3

3. The use of a common reference at Cz might be additionally problematic, and can pose a known problem for PLV analyses.

As the referee raised, PLV calculation for the signals derived from the same reference would inflate PLV values due to linear mixing in which the same source can contribute to both channels (Aydore et al. 2013). However, we believe the problem of common reference was addressed by re-reference of EEG signal with the bilateral Laplacian montage for online and common-average reference for offline sensor-space analysis and calculated M1-M1 PLV. We clarified this in the method section as follows:

[Line 688] In the state-dependent sessions, the measured EEG signals were subjected to custom MATLAB scripts and processed in the following manner; the bandpass filtered EEG signals around bilateral SM1 were processed with a Hjorth-style Laplacian spatial filter and the filter calibrated for FOI of each participant.

[Line 715] Then, the EEG signals processed with common average reference were subjected to cortical source estimation using Brainstorm toolbox(Tadel et al. 2011) with the normalized brain and sLoreta algorithm(Pascual-Marqui 2002).

Comment 1.4

4. For their offline analyses, the authors assess activity in M1 and SMA in source space. This does not mean that volume conduction can be ruled out however. However, the paper is relatively sparse on detail for this analysis and the results. It would be reassuring to see the source data in a lot more detail. Put simply, the proximity of the three brain regions of interest make signal leakage a real concern. At times it is also not clear whether the reported results are all based on the offline source data, or whether some results are derived from sensor space analyses.

First, we visualized the representative data of sensor-space and source-space signals derived from scalp EEG signals in a randomly chosen trial as shown in Fig. R6

Fig. R6. Examples of raw EEG signals derived from a single trial

a Beta-band EEG signals in sensor-space. As representatives of sensor-space signals, those derived from FCz, C3 and C4 were shown. The top three panels indicate whole-trial signals filtered by the narrow band-pass filter ($IBF \pm 1\text{Hz}$) and the bottom three panels indicate those during the ready and movement onset period. **b** Beta-band EEG signals in source-space. As representatives of source-space signals, those derived from SMA and bilateral M1 were shown. **c** Alpha-band EEG signals in sensor-space. The trial identical to (a) and (b) are shown. The top three panels indicate whole-trial signals filtered by the narrow band-pass filter ($IAF \pm 1\text{Hz}$) and the bottom three panels indicate those during the ready and movement onset period. **d** Alpha-band EEG signals in source-space. As representatives of source-space signals, those derived from SMA and bilateral M1 were shown.

Qualitatively, the sensor-space signals (especially those in alpha-band) indicate similar waveforms, putatively due to the zero-lag correlation induced by the volume conduction. Meanwhile, the sensor-space signals exhibit the alleviated patterns across three regions. We investigated the difference in the effects of zero-lag correlation between sensor- and source-space data using the Pearson’s correlation coefficient as the metric of potential signal leakage. As shown in Fig. R7, the source-space signals significantly lower amplitude correlation compared to the sensor-space signals, consistent with the qualitative observation for

COMMSBIO-22-2454R1: Beta rhythmicity in human motor cortex reflects neural population coupling that modulates subsequent finger coordination stability

the single trial data. (Paired t -test, beta: $t(14) = 4.11$, $p = 0.001$, $d = 1.06$, $CI_{95} = [0.41 \ 1.69]$, alpha: $t(14) = 5.43$, $p < 0.001$, $d = 1.40$, $CI_{95} = [0.67 \ 2.11]$).

Fig. R7. Comparison of zero-lag correlation coefficients between sensor- and source-space EEG signals.

Since our analysis successfully captured the relationship between behavioral performance and M1-M1 phase coupling in both source and sensor-space in the beta-band signals, we believe the stronger volume conduction in the sensor-space does not critically change our interpretation of main results, we added the corresponding discussion in the revised manuscript as follows:

[Line 471] Although the Hjorth montage extracts local neural oscillations under the electrode (Hjorth 1975; Stefanou et al. 2018; Tsuchimoto et al. 2021), the possibility that neural oscillations outside M1 or **zero-lag phase synchrony due to volume conduction of EEG** influence online PLV estimation cannot be excluded. In addition, to clarify whether analyses were based on the sensor or source-space data, we labeled each panel as “sensor” or “source” in the revised manuscript.

Comment 1.5

5. One way to address these concerns is to employ surrogate data and analyses, and one would hope to see these for both alpha and beta.

According to the referee’s suggestion, we addressed the concern regarding spatial leakage by surrogate data analysis. We generated phase-shuffled surrogate dataset and compared the PLV distribution with the original condition. As shown in the Supplementary Fig. 1c, the distribution of PLV was significantly altered after the permutation (Grand-average of surrogate data were shown). The null distribution of PLV values indicate the original difference was significant in the permutation test (Beta: $p < 0.05$, Alpha: $p < 0.05$).

Fig. R8 Comparison of M1-M1 PLV between actual and phase-shuffled surrogate dataset.

The supplementary analysis was implemented in the revised manuscript and supplementary information as follows:

[Line 184] We found similar results using alternative parameters and metrics for phase coupling applied to the source-space EEG data (Supplementary Fig. 3 and 4a-c), **but not for phase-shuffled surrogate dataset (Supplementary Fig. 4d).**

[Line 64 in SI] **d Comparison of M1-M1 PLV derived from actual and phase-shuffled surrogate data. The distribution of PLV was significantly altered after the permutation (Grand-average of surrogate data were shown). The null distribution of PLV values indicate the original difference was significant in the permutation test (Beta: $p < 0.05$, Alpha: $p < 0.05$).**

Comment 1.6

The paper is somewhat vague on the biological mechanism at hand. In essence, it relies on a rather vague notion of pre-movement ‘interactions’. Some questions come to mind: what is the specific role of beta (vs other frequencies) in pre-movement activity? The suggestions that “beta-dominant findings suggest that cortical pre-movement interaction organizes motor network to support subsequent stability of bimanual coordination” is rather vague and hard to falsify. How would, for example, a change in beta in this network before the onset of bi-manual coordinated movement – lasting 10s of seconds as here – provide a behavioural advantage? There is an obvious reason for focusing on pre-movement activity, as this is uncontaminated by overt movement, but it raises the question how that activity influences an nth movement occurring a relatively long time later? More broadly, what is happening physiologically prior to movement that leads to (if true) changes in bimanual behavior? Related, what are the physiological mechanisms that lead to PLV increases and how would that potentially contribute to later performance changes? I appreciate the answers may not be known; a slightly more concrete interpretation of how a signal in a network at one point (pre movement) can lead to the behavioural changes seen 10s of seconds

later would seem relevant.

We thank the referee for allowing us to clarify the point. Please find our responses to the referee's questions raised in the comment below.

(1) What is the specific role of beta (vs other frequencies) in pre-movement activity?

A variety of studies have demonstrated that beta-band activities in the preparatory period of discrete motor tasks such as the simple reaction task or stop reaction task is associated with the subsequent performance (e.g., reaction time)(He et al. 2020; Khanna and Carmena 2017). It is because the beta-band activity generated by basal ganglia-thalamo-cortical loop, that is involved in the excitatory and inhibitory control of motor cortical excitability influence the motor output efficacy (Diesburg et al. 2021; Schilberg et al. 2018; Torrecillos et al. 2020). Meanwhile, oscillatory activities in other frequency bands such as theta (3-7 Hz), alpha (8-13 Hz) in M1 are likely implicated in the other cortical computation such as action selection (Brinkman et al. 2014; Denis et al. 2017), and coupled with gamma (30- Hz) activities reflecting neuronal firing (Yanagisawa et al. 2012). Hence, preparatory beta-band activities may play a pivotal role in controlling corticospinal output.

(2) How would, for example, a change in beta in this network before the onset of bi-manual coordinated movement – lasting 10s of seconds as here – provide a behavioural advantage?

Since the pre-movement activities of both M1 and SMA play significant roles to characterize subsequent behavioral parameters, likely independently from the output-related activities (Kaufman et al. 2014; Tanji and Shima 1994), the preparatory activities would lead the motor network to the suitable state for the subsequent movement. Especially at the level of macroscopic neural systems, since the neural oscillations derived from the multiple regions are thought to emerge from system with coupled oscillators (Cabral et al. 2022; Onojima et al. 2018), the pre-movement M1-M1 phase coupling would determine the initial state of motor network which influences the temporal development after movement onset.

(3) There is an obvious reason for focusing on premovement activity, as this is uncontaminated by overt movement, but it raises the question how that activity influences an nth movement occurring a relatively long time later?

As the referee pointed out, it is the most remarkable finding in this study that specific pre-movement cortical activity patterns influence an nth movement occurring a relatively long time later. Compared to the relationship between the performance of discrete motor tasks, it remains unclear whether the beta-band activity has lasting effects on the ability to maintain the instable behavioral pattern (e.g., anti-phase bimanual coordination). We speculate the mechanism behind the relationship is the pre-movement activities which lead bilateral M1 to the action-specific state determines the initial state of motor network and interference of bilateral corticomotor representation, which emerges through consolidation of acquired

motor program.

Since the beta-band oscillations derived from scalp EEG signals represent motor cortical activities which regulates corticomotor output (Espenhahn et al. 2019; Omlor et al. 2011; Rueda-Delgado et al. 2017), the cortico-cortical communication tested with the M1-M1 phase coupling in the beta-band at pre-movement period would reflect the initial state of motor network, which influences subsequent motor performance (duration of finger tapping) putatively not by directly modulating the transition-related activity separated in time, but by attenuating the interference of prepared motor program ideally distinctly represented in bilateral M1 to sustain the simultaneous activation of left index and right middle finger muscles and vice versa (Houweling et al. 2010).

(4) More broadly, what is happening physiologically prior to movement that leads to (if true) changes in bimanual behavior? Related, what are the physiological mechanisms that lead to PLV increases and how would that potentially contribute to later performance changes?

Not only in the motor domain, neural processes in cognition and perception are dynamically modulated by the neural oscillations prior to physiological events (e.g., stimulus presentation) (Bagherzadeh et al. 2020; Romei et al. 2008, 2010). In general, we believe ongoing neural oscillations represent cortical dynamics, which relates to subsequent states of neural systems and the corresponding behavior. Hence, during the motor preparation of bimanual finger tapping, bilateral M1 are brought to the suitable state for the subsequent behavior putatively by the modulatory input from higher motor cortices such as SMA, as suggested by the significant decrease in pre-movement M1-M1 PLV in the sham group of experiment 2.

Indeed, the effects of interaction between the M1-M1 PLV and SMA on the duration of anti-phase bimanual tapping were dominantly observed after the short-term break. It should be because the acquisition of motor program, which is the modulatory activities from higher-motor cortex (Perez et al. 2008; Sun et al. 2022; Tanaka et al. 2010), requires consolidation to make activities during preparatory period possible to influence the subsequent performance. Moreover, the result that rTMS treatment group did not exhibit the performance improvement suggests the acquired preparatory activity patterns (lower M1-M1 PLV and SMA spectral power) is necessary to achieve sustained anti-phase bimanual tapping.

I appreciate the answers may not be known; a slightly more concrete interpretation of how a signal in a network at one point (pre movement) can lead to the behavioural changes seen 10s of seconds later would seem relevant.

Although the explanation above is speculative due to the insufficient exploration of continuous motor tasks combined with populational neural measurements to quantify the macroscopic neural coupling. We employed the explanation in the revised manuscript so that the hypothesis can be investigated in the further studies.

[Line 434] Specifically, the pre-movement activities which lead bilateral M1 to the action-specific state would determine the initial state of the motor network and interference of bilateral corticomotor representation, which emerges through consolidation of acquired motor program^{50,73}. Since the beta-band oscillations derived from scalp EEG signals represent motor cortical activities that regulate corticomotor output⁷⁴⁻⁷⁶, the cortico-cortical communication tested with the M1-M1 phase coupling in the beta-band at the pre-movement period would reflect the initial state of motor network, which influences subsequent motor performance (duration of finger tapping) putatively not by directly modulating the transition-related activity separated in time, but by attenuating the interference of prepared motor program ideally distinctly represented in bilateral M1 to sustain the simultaneous activation of left index and right middle finger muscles and *vice versa*²⁴. Indeed, the effects of interaction between the M1-M1 PLV and SMA on the duration of anti-phase bimanual tapping were dominantly observed after short-term break. It should be because the acquisition of motor program, which is the modulatory activities from higher-motor cortices^{40, 41, 48} requires consolidation to make activities during preparatory period possible to influence the subsequent performance. Moreover, the result that rTMS treatment group did not exhibit the performance improvement suggests the acquired preparatory activity patterns (lower M1-M1 PLV and SMA spectral power) is necessary to achieve sustained anti-phase bimanual tapping.

Other comments:

Comment 1.7

I found the paper overall accessible and nice to read but encourage additional proof reading for minor grammatical errors.

According to the suggestion, we used the additional proof-reading service on the revised manuscript. We corrected the manuscript throughout.

Comment 1.8

On line 273, it seems that an interaction in the 2-way ANOVA was followed up with post-hoc ttests. To the best of my knowledge, this is not the approach for dealing with interactions. Please check your analyses throughout.

Thank you for allowing us to clarify this point. We specifically used a mixed two-way repeated measures ANOVA with one within-subjects factor (Time: pre and post-evaluation) and one between-subjects factor (Group: treatment and sham). We believe this is the appropriate methods to evaluate the interaction effect such as (Biasiucci et al. 2018). The expression of employed statistical methods was rephrased as follows:

[Line 270] (Fig. 4b, a mixed two-way repeated measures ANOVA, $F(1, 28) = 4.94, p = 0.035, \eta^2 = 0.077,$

COMMSBIO-22-2454R1: Beta rhythmicity in human motor cortex reflects neural population coupling that modulates subsequent finger coordination stability

main effect of “Time”, $F(1, 28) = 6.73$, $p = 0.015$, $\eta^2 = 0.11$), but not a main effect of “Group” ($F(1, 28) = 1.18$, $p = 0.29$).

Comment 1.9

The data sharing statement seems rather weak and untimely. For a publication of this kind one might expect a lot more transparency and openness.

According to the suggestion, we uploaded the anonymized experiment data on the file sharing service with analysis code.

https://figshare.com/projects/Beta_rhythmicity_in_human_motor_cortex_reflects_neural_population_coupling_that_modulates_subsequent_finger_coordination_stability/144057

Accordingly, the data availability statement was revised.

[Line 1029] ~~The data that support the findings of this study are available from the corresponding author upon reasonable request.~~ All data supporting the findings of the study are available from figshare (https://figshare.com/projects/Beta_rhythmicity_in_human_motor_cortex_reflects_neural_population_coupling_that_modulates_subsequent_finger_coordination_stability/144057).

References in response to referees

Ariani G, Pruszynski JA, Diedrichsen J. Motor planning brings human primary somatosensory cortex into action-specific preparatory states. *Elife* 11, 2022.

Aydore S, Pantazis D, Leahy RM. A note on the phase locking value and its properties. *Neuroimage* 74: 231–244, 2013.

Bagherzadeh Y, Baldauf D, Pantazis D, Desimone R. Alpha Synchrony and the Neurofeedback Control of Spatial Attention. *Neuron* 105: 577-587.e5, 2020.

Biasiucci A, Leeb R, Iturrate I, Perdakis S, Al-Khodairy A, Corbet T, Schnider A, Schmidlin T, Zhang H, Bassolino M, Viceic D, Vuadens P, Guggisberg AG, Millán JDR. Brain-actuated functional electrical stimulation elicits lasting arm motor recovery after stroke. *Nat Commun* 2018 9: 1–13, 2018.

Brinkman L, Stolk A, Dijkerman HC, De Lange FP, Toni I. Distinct Roles for Alpha- and Beta-Band Oscillations during Mental Simulation of Goal-Directed Actions. *J Neurosci* 34: 14783–14792, 2014.

Bruña R, Maestú F, Pereda E. Phase locking value revisited: teaching new tricks to an old dog. *J Neural Eng* 15: 056011, 2018.

Cabral J, Castaldo F, Vohryzek J, Litvak V, Bick C, Lambiotte R, Friston K, Kringelbach ML, Deco G. Metastable oscillatory modes emerge from synchronization in the brain spacetime connectome. *Commun Phys* 2022 5: 1–13, 2022.

Denis D, Rowe R, Williams AM, Milne E. The role of cortical sensorimotor oscillations in action anticipation. *Neuroimage*

COMMSBIO-22-2454R1: Beta rhythmicity in human motor cortex reflects neural population coupling that modulates subsequent finger coordination stability

146: 1102–1114, 2017.

Diesburg DA, Greenlee JDW, Wessel JR. Cortico-subcortical β burst dynamics underlying movement cancellation in humans. *Elife* 10, 2021.

Espenhahn S, van Wijk BCM, Rossiter HE, de Berker AO, Redman ND, Rondina J, Diedrichsen J, Ward NS. Cortical beta oscillations are associated with motor performance following visuomotor learning. *Neuroimage* 195: 340–353, 2019.

He S, Everest-Phillips C, Clouter A, Brown P, Tan H. Neurofeedback-linked suppression of cortical B bursts speeds up movement initiation in healthy motor control: A double-blind sham-controlled study. *J Neurosci* 40: 4021–4032, 2020.

Houweling S, Beek PJ, Daffertshofer A. Spectral changes of interhemispheric crosstalk during movement instabilities. *Cereb Cortex* 20: 2605–2613, 2010.

Kaufman MT, Churchland MM, Ryu SI, Shenoy K V. Cortical activity in the null space: Permitting preparation without movement. *Nat Neurosci* 17: 440–448, 2014.

Khanna P, Carmena JM. Beta band oscillations in motor cortex reflect neural population signals that delay movement onset. *Elife* 6, 2017.

Omlor W, Patino L, Mendez-Balbuena I, Schulte-Mönting J, Kristeva R. Corticospinal beta-range coherence is highly dependent on the pre-stationary motor state. *J Neurosci* 31: 8037–8045, 2011.

Onojima T, Goto T, Mizuhara H, Aoyagi T. A dynamical systems approach for estimating phase interactions between rhythms of different frequencies from experimental data. *PLOS Comput Biol* 14: e1005928, 2018.

Pascual-Marqui RD. Standardized low-resolution brain electromagnetic tomography (sLORETA): Technical details. In: *Methods and Findings in Experimental and Clinical Pharmacology*. 2002, p. 5–12.

Perez MA, Tanaka S, Wise SP, Willingham DT, Cohen LG. Time-Specific Contribution of the Supplementary Motor Area to Intermanual Transfer of Procedural Knowledge. *J Neurosci* 28: 9664–9669, 2008.

Romei V, Brodbeck V, Michel C, Amedi A, Pascual-Leone A, Thut G. Spontaneous fluctuations in posterior α -band EEG activity reflect variability in excitability of human visual areas. *Cereb Cortex* 18: 2010–2018, 2008.

Romei V, Gross J, Thut G. On the role of prestimulus alpha rhythms over occipito-parietal areas in visual input regulation: Correlation or causation? *J Neurosci* 30: 8692–8697, 2010.

Rueda-Delgado LM, Solesio-Jofre E, Mantini D, Dupont P, Daffertshofer A, Swinnen SP. Coordinative task difficulty and behavioural errors are associated with increased long-range beta band synchronization. *Neuroimage* 146: 883–893, 2017.

Schilberg L, Engelen T, ten Oever S, Schuhmann T, de Gelder B, de Graaf TA, Sack AT. Phase of beta-frequency tACS over primary motor cortex modulates corticospinal excitability. *Cortex* 103: 142–152, 2018.

Sun X, O’Shea DJ, Golub MD, Trautmann EM, Vyas S, Ryu SI, Shenoy K V. Cortical preparatory activity indexes learned motor memories. *Nature* 602: 274–279, 2022.

Tanaka S, Honda M, Hanakawa T, Cohen LG. Differential contribution of the supplementary motor area to stabilization of a procedural motor skill acquired through different practice schedules. *Cereb Cortex* 20: 2114–2121, 2010.

COMMSBIO-22-2454R1: Beta rhythmicity in human motor cortex reflects neural population coupling that modulates subsequent finger coordination stability

Tanji J, Shima K. Role for supplementary motor area cells in planning several movements ahead. *Nature* 371: 413–416, 1994.

Torrecillos F, Falato E, Pogosyan A, West T, Di Lazzaro V, Brown P. Motor cortex inputs at the optimum phase of beta cortical oscillations undergo more rapid and less variable corticospinal propagation. *J Neurosci* 40: 369–381, 2020.

Tsuchimoto S, Shibusawa S, Iwama S, Hayashi M, Okuyama K, Mizuguchi N, Kato K, Ushiba J. Use of common average reference and large-Laplacian spatial-filters enhances EEG signal-to-noise ratios in intrinsic sensorimotor activity. *J Neurosci Methods* 353, 2021.

Yanagisawa T, Yamashita O, Hirata M, Kishima H, Saitoh Y, Goto T, Yoshimine T, Kamitani Y. Regulation of motor representation by phase-amplitude coupling in the sensorimotor cortex. *J Neurosci* 32: 15467–15475, 2012.

Referee #2:

General remark

This is the study investigating whether pre-movement cortical interaction in the motor network modulates subsequent bimanual coordination performance. The authors reported the negative correlation between the duration of anti-phase bimanual tapping and pre-movement M1-M1 PLV, and the significant effects of M1-M1 PLV and ERSP at the SMA on the duration. Further, they reported that the effects of the interaction and performance improvement were disrupted by low-frequency rTMS over the SMA.

This study provides an important contribution to the understanding of the effects of the dynamics of pre-movement neural oscillations in the motor network on bimanual coordination. However, a revision is needed before it can be accepted for publication.

We thank for constructive and insightful comments for our manuscript. We revised our manuscript according to comments. Please find the point-by-point responses to your comments.

Major comments:

Comment 2.1

This study investigated anti-phase bimanual coordination stability after consolidation in Experiment 1 and perturbed the consolidation in Experiment 2. The findings about the correlation among the duration of anti-phase bimanual tapping, pre-movement M1-M1 PLV, and ERSP at the SMA were limited to the state after the consolidation. Actually the authors did not find the significant correlation among them in Pre-stimulation in Experiment 2 (Supplementary Fig. 4). When they described the findings in Abstract and Discussion, they should refer to the limitation.

The referee is right. We revised the sections as follows:

[Line 38] Our results demonstrate that pre-movement cortical oscillatory coupling within the motor network unknowingly influences bimanual coordination performance in humans **after consolidation**.

[Line 371] The interaction of M1-M1 PLV and ERSP magnitude of SMA explained the inter-individual variability of duration **after consolidation of finger tapping task**.

[Line 377] We found that their oscillatory activities during motor preparation organize the motor network for subsequent performance improvement **after consolidation**,

[Line 491] In summary, we showed that the manipulation of **movement onset based on** pre-movement intrinsic brain rhythmicity improved the ability to maintain anti-phase bimanual coordination **after consolidation**.

Comment 2.2

I think correlation analysis and multiple regression analysis in this study should be performed separately for the data before the consolidation and those after the consolidation.

The results of experiment 1 employed analysis for exclusively on the data after the short-term consolidation. For the results of experiment 2, we used both pre-and post-evaluation data to account for the difference in the conditions. However, given that the pre-evaluation data is unnecessary to corroborate the main findings in the study, we reanalyzed data using only the post-data.

Fig. 4 Treatment effects of repetitive transcranial magnetic stimulation (rTMS)

[Line 348] e Linear-mixed model for the performance and pre-movement cortical activity patterns in beta-band. In line with the data from experiment 1 (Fig. 3), a significant main effect was observed for M1-M1 PLV and ERSP magnitude at SMA, and their interaction revealed significant interaction of SMA-ERSP × M1-M1 PLV effect only in sham treatment group (treatment: ERSP: $p = 0.0036$ $p = 0.01$, PLV: $p = 0.244$ $p < 0.001$, ERSP × PLV: $p = 0.12$ $p = 0.72$; Sham: ERSP: $p < 0.004$ $p = 0.038$, PLV: $p = 0.0034$ $p = 0.04$, ERSP × PLV: $p < 0.004$ $p = 0.03$).

[Line 321] A linear-mixed model based on ERSP at SMA, M1-M1 PLV, and their interactions revealed that the interaction with M1-M1 PLV and ERSP at SMA were only significant in the sham group but not in the treatment group (Fig. 4e: treatment: ERSP, $p = 0.0036$ $p = 0.01$, PLV: $p = 0.244$ $p < 0.001$, ERSP × PLV: $p = 0.12$ $p = 0.72$; Sham: ERSP: $p < 0.004$ $p = 0.038$, PLV: $p = 0.0034$ $p = 0.04$, ERSP × PLV: $p < 0.004$ $p = 0.03$),

COMMSBIO-22-2454R1: Beta rhythmicity in human motor cortex reflects neural population coupling that modulates subsequent finger coordination stability

[Line 777] For multiple regression analysis using ERSP magnitude at SMA and M1-M1 PLV as explanatory variables, mixed-effect models with normal distribution as the link function for post-evaluation data **to consider the tapping frequency as a random effect**⁹⁶. ~~and participant as a random effect were used in the data from experiment 2, since multiple data from a single participant (i.e., pre and post-evaluation data) were included~~

Comment 2.3

I found the descriptions such as ‘manipulation of pre-movement M1-M1 PLV’ in the manuscript. However, the authors did not manipulate the PLV, but they manipulated the timing of the movement cues depending on the PLV. They should revise them.

The referee is right. We did not manipulate the PLV itself. We rephrased the corresponding description as follows:

[Line 39] Our results demonstrate that pre-movement cortical oscillatory coupling within the motor network unknowingly influences bimanual coordination performance in humans, suggesting feasibility of augmenting human motor ability by covertly ~~manipulating~~ **monitoring** preparatory neural dynamics.

[Line 65] However, there is no direct evidence that manipulation of **movement onset dependent on** fluctuating cortical coupling among bilateral M1 and SMA modulates subsequent performance of bimanual coordination.

[Line 71] Herein, we hypothesized that manipulating **movement cues based** on pre-movement phase coupling of bi-hemispheric SMR signals might influence stability of subsequent anti-phase bimanual coordination,

[Line 77] the manipulation of **movement cues based on** pre-movement M1-M1 phase synchrony modulates the duration of anti-phase bimanual tapping (Fig. 1b).

[Line 82] Experiment 1, which used brain state-dependent movement initiation, revealed whether the manipulation of **movement cues based on** pre-movement M1-M1 phase synchrony improves motor performance.

[Line 83] Manipulation of **movement cues based on** pre-movement phase synchrony

[Line 154] manipulation of **movement cues based on** pre-movement M1-M1 PLV modulated subsequent stability of anti-phase bimanual coordination performance.

[Line 475] Moreover, although a significant difference in within-participant performance was demonstrated by M1-M1 PLV-**based movement cueing** ~~manipulation~~,

[Line 491] In summary, we showed that the manipulation of **movement onset based on** pre-movement intrinsic brain rhythmicity improved the ability to maintain anti-phase bimanual coordination **after**

COMMSBIO-22-2454R1: Beta rhythmicity in human motor cortex reflects neural population coupling that modulates subsequent finger coordination stability

consolidation.

[Line 496] suggesting the feasibility of augmenting human motor ability by its covert manipulation control of movement cues based on the macroscopic neural coupling.

Comment 2.4

Some participants showed higher M1-M1 PLV in PLV Low condition than in PLV High condition (Fig. 2a). Were these participants included in further analysis? I think they should be excluded from further analysis.

The participants who did not exhibit successful PLV control did not satisfy the inclusion criterion. Therefore, the original results did not include the participants. The point was explicitly described as follows:

[Line 163] Note that only participants who contained more than 5 trials for each condition and whose pre-movement PLV was successfully conditioned were included in the statistical test (See material and methods for details).

Minor comments:

Comment 2.5

Fig.1

c Top: What does the yellow bar represent?

e: The authors should describe what the black lines represent (the same in other figures). The black lines were missing in Session 4 and 5.

The yellow bar in Fig. 1c indicates ready period and the black line in Fig.1 e represents mean values. We added the description in the figure legend and revised the figure as follows:

Fig. 1 Anti-phase bimanual tapping task

[Line 141] The yellow area indicates the ready period whose duration was variable

Comment 2.6

Page 8 Line 161

The authors should explain the rationale for using 5 trials as the inclusion criteria.

During the preliminary experiments employing the anti-phase tapping task, the performance measured by its duration was variable within participants. To effectively capture the performance of each participant in each condition, we determined the threshold of inclusion criteria based on the aggregated dataset of preliminary experiments in which participants experienced at least 15 trials at their optimal frequencies. The Figure RX indicates the average duration of data subsampled from the whole data (x-axis) for each participant (Shades indicate the standard error computed using the iteratively subsampled dataset), and the deviation from grand average (y-axis), computed using the whole data. We set the threshold which exhibited 1.96 SE (the standard error of the difference from the grand average) is less than 2s (10% of the task duration). Given that 4.7 samples were necessary to calculate reliable average duration, we determined the 5 trials as the inclusion criteria for each analysis. The procedure to determine the inclusion criteria was described in the method section as follows:

Supplementary Fig. 11. Variance of anti-phase duration for dataset of preliminary experiments

[Line 752] The inclusion criterion was determined based on results of a preliminary experiment to quantify the trial-to-trial variability of anti-phase tapping performance (Supplementary Fig. 11).

In the supplementary information (SI), we added the figure above. The figure legend was described as follows:

[Line 155 in SI] Supplementary Fig. 11: Variance of anti-phase duration for dataset of preliminary experiments.

Each line indicates the average duration of data subsampled from the whole data (x-axis) for each participant (Shades indicate the standard error computed using the iteratively subsampled dataset), and the deviation from grand average (y-axis), computed using the whole data. We set the threshold which exhibited 1.96 SE (the standard error of the difference from the grand average) is less than 2s (10% of the task duration). As indicated in the black line, 4.7 trials were necessary.

Comment 2.7

Fig.3

Are the data in Fig. 3 those from Session 2 – 5? I think they should not include those from Session 1. (Please see above)

The reviewer is right. Only data from 2-5 sessions were subjected to the analysis in Fig. 3. We clarified the point as follows:

[Line 760] For each participant, averaged data were calculated from trials in state-dependent conditions (i.e., sessions 2-5), which did not begin based on M1-M1 PLV.

Comment 2.8

Fig. 3 c

What does the blue curved surface represent?

The surface is regression plane of multiple regression model. The legend was elaborated as follows:

[Line 243] The blue curved surface represents the regression model.

Comment 2.9

Fig. 4

There is no description about which is pre- or post-evaluation (the same in other figures).

We added ticks indicating pre-and pos-evaluation in each figure in the revised manuscript. Please refer Fig. 4, Supplementary Fig. 3, 5 and 7.

Comment 2.10

Fig. 4d and e

Are the data in Fig. 4d and e those from post-evaluation? I think they should not include those from pre-evaluation. (Please see above)

Fig. 4d is the analysis for changes in the ERSP magnitude and duration from pre- to post-evaluation sessions. For Fig. 4e, we conducted reanalysis with the data of post-evaluation and found similar results.

Comment 2.11

Fig. 4e

What do the green and orange curved surfaces represent?

The surfaces are regression plane of multiple regression model. The legend was elaborated as follows:

[Line 353] The green and orange curved surfaces represent regression model derived from the treatment and sham groups, respectively.

Comment 2.12

Page 30

The authors should add description about how the responsive frequency and the COIs were determined. Was the responsive frequency the one showing maximum ERSP? Were the COI the one showing maximum median values of ERSP?

We thank the referee for allowing us to clarify this point. For each frequency band (alpha and beta), the averaged magnitude of ERSP in were calculated in 8-13 Hz for alpha and 14-30 Hz at 3 Hz sliding bin and 2 Hz overlap. Then, the frequency of interest was determined by those that exhibited minima of ERSP magnitude in C3 and C4 channels (averaged across time at task period and channels) since the power attenuation of sensorimotor rhythm (SMR) around bilateral M1 reflects corticospinal excitability (Hummel et al 2002, Takemi et al., 2013; 2018). Then, the COI was determined from the channels which exhibited

COMMSBIO-22-2454R1: Beta rhythmicity in human motor cortex reflects neural population coupling that modulates subsequent finger coordination stability

the most prominent SMR-ERD from the electrodes around bilateral M1. The additional description was added in the revised manuscript as follows:

[Line 672] For each frequency band (alpha and beta), the averaged magnitude of ERSP in were calculated in 8-13 Hz for the alpha and 14-30 Hz for the beta at 3 Hz sliding bin and 2 Hz overlap. The frequency of interest (FOI) was determined by those that exhibited minima of ERSP magnitude in C3 and C4 channels (averaged across time at task period and channels) since the power attenuation of SMR around bilateral M1 reflects corticospinal excitability⁹¹⁻⁹³. Then, the topographic representation of the frequency-of-interest (FOI) was visualized.

Comment 2.13

Page 30

Did the authors use all the EEG data (15 min) for calibration?

Please note that we measured 15-s resting EEG data at the beginning of each session, and used all 15-s data to calculate the threshold for PLV-triggered movement cueing. We elaborated our manuscript to clarify the point as follows.

[Line 699] To determine the thresholds at the beginning of each session, participants underwent a 15-s rest. After outlier detection using median absolute deviation for all data samples (MAD, points deviating more than 3 times the MAD were rejected), 25% and 75% quartiles of PLV distribution were set as the threshold for low and high PLV states, respectively.

Comment 2.14

Supplementary Fig. 1f

Please explain why the lines representing PLV Low condition don't have a negative peak at (or around) Time 0.

In the PLV high condition, the PLV values exhibited consisted increase from resting to ready period. Meanwhile, in PLV low condition possibly two directions of PLV changes were detected: the trend from lower to higher PLV (rising flanks) and from higher to lower (falling flanks). The two conditions were not discriminated in the current experimental paradigm, since only the moving averaged current PLV was considered to trigger movement cues. Hence the two anti-phase time-course of PLV values in the PLV-Low condition may cancelled out and no negative peaks were found. This explanation was added in the figure legend of Supplementary Fig. 1f as follows:

[Line 19 in SI] Since the online PLV-triggered algorithm only uses the ready period data, the trend of PLV was not considered and a negative peak in PLV low condition was not found due to the cancellation of

rising and falling flanks.

Comment 2.15

Supplementary Fig. 7

I think the data in Supplementary Fig. 7 should not include those from pre-evaluation. (Please see above)

According to the suggestion, we conducted reanalysis with the data of post-evaluation and found similar results. The results were implemented in the revised manuscript as follows:

Supplementary Fig. 10. Linear mixed-effect model considering cortical activity patterns in alpha-band

[Line 150 in SI] *Left*: Data from the treatment group (ERSP × PLV: $p = 0.74$ **0.68**; ERSP: $p = 0.025$ **0.86**, M1-M1 PLV: $p = 0.474$ **0.97**). *Right*: Data from the sham group (ERSP × PLV: $p = 0.78$ **0.14**; ERSP: $p = 0.39$ **0.50**, M1-M1 PLV: $p = 0.45$ **0.90**)

Comment 2.16

Typo

Page 14 Line 285

0.85. -> 0.85).

Page 15 Line 302

group Fig. 4d -> group (Fig. 4d

We thank the referee for the careful inspection of our manuscript. We revised them in the revised manuscript:

References in response to referees

Hummel F, Andres F, Altenmüller E, Dichgans J, Gerloff C. Inhibitory control of acquired motor programmes in the human brain. *Brain* 125: 404–420, 2002.

Takemi M, Maeda T, Masakado Y, Siebner HR, Ushiba J. Muscle-selective disinhibition of corticomotor representations

COMMSBIO-22-2454R1: Beta rhythmicity in human motor cortex reflects neural population coupling that modulates subsequent finger coordination stability

using a motor imagery-based brain-computer interface. *Neuroimage* 183: 597–605, 2018.

Takemi M, Masakado Y, Liu M, Ushiba J. Event-related desynchronization reflects downregulation of intracortical inhibition in human primary motor cortex. *J Neurophysiol* 110: 1158–1166, 2013.

REVIEWERS' COMMENTS:

Reviewer #1 (Remarks to the Author):

The authors have thoroughly revised their manuscript including new control analyses to all the points raised in my previous comments. Overall these analyses complement the original results. The discussion was similarly revised. The authors have done everything asked from them, with thanks.

Reviewer #4 (Remarks to the Author):

The authors have provided a thorough revision of the manuscript which has been substantially improved.

I still have one question. In Supplementary Figure 11, is the unit of y-axis second?

COMMSBIO-22-2454A2: Beta rhythmicity in human motor cortex reflects neural population coupling that modulates subsequent finger coordination stability

Point-by-point answers to comments from referees

We thank the referees and editor for their support and constructive suggestions. We addressed all concerns of the reviewers in our updated manuscript. Please find detailed explanations below.

Referee #4:

The authors have provided a thorough revision of the manuscript which has been substantially improved. I still have one question. In Supplementary Figure 11, is the unit of y-axis second?

Yes, the unit is second (as specified in the y-axis of the figure). We put the note in the figure caption as follows:

[Line 157] Each line indicates the average duration of data subsampled from the whole data (x-axis) for each participant (Shades indicate the standard error computed using the iteratively subsampled dataset), and the deviation from the grand average (**second**, y-axis), computed using the whole data.